# Recognition and coacervation of G-quadruplexes by a multifunctional disordered region in RECQ4 helicase

Anna C. Papageorgiou[1,9], Michaela Pospisilova[2,3,9], Jakub Cibulka [3], Raghib Ashraf[2], Christopher A. Waudby [4,5], Pavel Kadeřávek [1], Volha Maroz[2,3], Karel Kubicek[1,2,6], Zbynek Prokop [7,8], Lumir Krejci [2,3,8] ✉ & Konstantinos Tripsianes [1] ✉

Biomolecular polyelectrolyte complexes can be formed between oppositely charged intrinsically disordered regions (IDRs) of proteins or between IDRs and nucleic acids. Highly charged IDRs are abundant in the nucleus, yet few have been functionally characterized. Here, we show that a positively charged IDR within the human ATP-dependent DNA helicase Q4 (RECQ4) forms coacervates with G-quadruplexes (G4s). We describe a three-step model of charge-driven coacervation by integrating equilibrium and kinetic binding data in a global numerical model. The oppositely charged IDR and G4 molecules form a complex in the solution that follows a rapid nucleation-growth mechanism leading to a dynamic equilibrium between dilute and condensed phases. We also discover a physical interaction with Replication Protein A (RPA) and demonstrate that the IDR can switch between the two extremes of the structural continuum of complexes. The structural, kinetic, and thermodynamic profile of its interactions revealed a dynamic disordered complex with nucleic acids and a static ordered complex with RPA protein. The two mutually exclusive binding modes suggest a regulatory role for the IDR in RECQ4 function by enabling molecular handoffs. Our study extends the functional repertoire of IDRs and demonstrates a role of polyelectrolyte complexes involved in G4 binding.

Guanine-rich nucleic acid sequences are prevalent in animal genomes and can spontaneously form G-quadruplexes (G4s) under physiological conditions[1]. These non-canonical secondary structures consist of repeating structural motifs called G-quartets that are stacked upon one another and are held together by an extensive network of hydrogen bonds linking four guanine bases around a cationic core[2-4]. G4s constitute a diverse family of highly stable DNA structures adopting parallel, hybrid or antiparallel topologies according to the pattern of the strand polarities and the orientation of the loops[2,5]. G4 sequences have been mapped within replication origins[6,7], mitochondrial DNA[8],

[1]CEITEC-Central European Institute of Technology, Masaryk University, Brno, Czech Republic. [2]National Centre for Biomolecular Research, Faculty of Science, Masaryk University, Brno, Czech Republic. [3]Department of Biology, Faculty of Medicine, Masaryk University, Brno, Czech Republic. [4]Institute of Structural and Molecular Biology, University College London, London WC1E 6BT, UK. [5]School of Pharmacy, University College London, London WC1N 1AX, UK. [6]Department of Condensed Matter Physics, Faculty of Science, Masaryk University, Brno, Czech Republic. [7]Loschmidt Laboratories, Department of Experimental Biology and RECETOX, Faculty of Science, Masaryk University, Brno, Czech Republic. [8]International Clinical Research Center, St Anne's University Hospital, Brno, Czech Republic. [9]These authors contributed equally: Anna C. Papageorgiou, Michaela Pospisilova. ✉e-mail: lkrejci@chemi.muni.cz; kostas.tripsianes@ceitec.muni.cz

telomeric ends[9], gene promoter regions[10,11], and oncogenes[12], and therefore have been implicated in a range of biological processes. Although G4s act as functional regulatory elements in different cellular contexts, they often need to be unfolded to prevent impediment of various DNA metabolic processes[13] such as DNA replication[14–17], epigenetic control[18–20], telomere homeostasis[21], and transcription[22].

DNA helicases in the RECQ[23–26], FANCJ[27–29], DEAH/RHA[30,31], and PIF1[32,33] families prevent G4-induced genome instability by selectively recognizing and unwinding G4s[34]. Through scarce structural studies and FRET-based kinetics, G4 unwinding models have been proposed for a subset of these helicases[35–38]. Aside from DNA helicases, replication protein A (RPA), the most abundant ssDNA binding protein, has also been implicated in destabilizing G4 structures[39–41]. In particular, RPA has been shown to facilitate the role of several helicases by direct protein-protein interactions or trapping unfolded G4s[17,42,43].

Over the past years, protein phase separation has been reported in many biological processes, and nucleic acids are widely involved in regulating biological condensates[44]. For example, DEAD-box RNA helicases, play critical roles in many aspects of RNA metabolism by undergoing liquid-liquid phase separation (LLPS) with their RNA targets and other proteins[45]. However, LLPS studies on DNA helicases are still minimal[46], and it is not known if the formation of biomolecular condensates could be involved in G4 metabolism.

Among the RECQ DNA helicases, RECQ4 has a unique domain organization[47] (Fig. 1a). It lacks the RecQ C-terminal (RQC) domain required for G4 unwinding by the other family members. On the other hand, it features an N-terminal region that includes segments homologous to the Sld2 protein, an essential DNA replication factor in deeper-branching eukaryotes[48–51]. This Sld2-like region (amino acids 1–400) of RECQ4 is important for DNA replication[52] and is indispensable for viability in metazoans[53–55]. Biochemical analysis has revealed multiple DNA binding sites in the Sld2-like region, which can recognize various DNA structures with a preference for Holliday junctions[56] and G4s[57]. It also provides a platform for many interacting partners, suggesting that the Sld2-like region is integral to RECQ4 functions in DNA replication[58–62], DNA repair[61,63], and mitochondrial maintenance[64]. However, the Sld2-like region is largely unstructured[57,65] (Supplementary Fig. 1), raising the question of what mechanistic/functional aspects may be associated with the evolutionary conserved prevalence of intrinsic disorder[66–68].

Owing to their conformational flexibility and dynamics, intrinsically disordered regions (IDRs) expand the biomolecular functionality of ordered proteins and domains[69–72]. They act as hubs mediating signals upon interaction with multiple targets[73]; they adopt different structures upon binding to different partners[74,75]; they participate in ultra-high affinity, extreme disorder[76], multivalent[77–79], or polyelectrolyte[80] interactions with the ability to drive liquid-liquid phase separation[81–83] and the assembly of biomolecular condensates[84]. The diverse functions of IDRs are related to information encoded in their sequences[85]. For instance, IDRs of DNA binding proteins are rich in positively charged residues that engage in DNA contacts[86]. However, it is unknown if such disordered cationic segments can have alternative functions other than DNA binding and consequently may regulate biological processes.

Here, we identify a positively charged RECQ4-specific motif (RSM) that interacts with RPA by undergoing a disorder-to-helix transition in a static complex with RPA protein. The intrinsic disorder allows RSM to also engage in polyelectrolyte interactions with oppositely charged DNA molecules. Interestingly, binding to G4 structures is followed by associative phase separation. The comprehensive kinetic analysis revealed that the coacervation of G4 and RSM follows a rapid nucleation-growth mechanism resulting in a dynamic equilibrium between dilute and condensed phases. RSM thus can switch between phases, conformational states and binding partners, providing a mechanism allowing diverse RECQ4 functions in the cell.

## Results

### A RECQ4-specific motif mediates an interaction with RPA

Members of the RECQ family are known to interact with RPA by utilizing unstructured motifs in their sequences[87–91]. As it has been shown that RECQ4 depletion in *Xenopus* extracts suppresses RPA loading at the origin of replication[52], we tested for a physical interaction between the two proteins in human cells. Using an anti-GFP antibody, we carried out co-immunoprecipitation (co-IP) experiments of EGFP-tagged RECQ4 in U2OS cells. In contrast to the EGFP control, EGFP-RECQ4-WT was able to pull-down RPA, suggesting a possible direct interaction between the two proteins (Fig. 1b).

We then sought to validate the interaction between RECQ4 and RPA in vitro. RPA is a protein composed of three subunits (RPA70, RPA32, RPA14) and exhibits a modular architecture[92,93] that allows globular domains to associate in a trimeric core (70C, 32D, 14), bind ssDNA (70A, 70B, 70C, 32D), or participate in protein interactions (RPA70N, RPA32C) (Supplementary Fig. 2). Since RPA mainly associates with unstructured motifs present in BLM[88,90], WRN[87], and other binding partners[94–97], we reasoned that the largely disordered Sld2-like

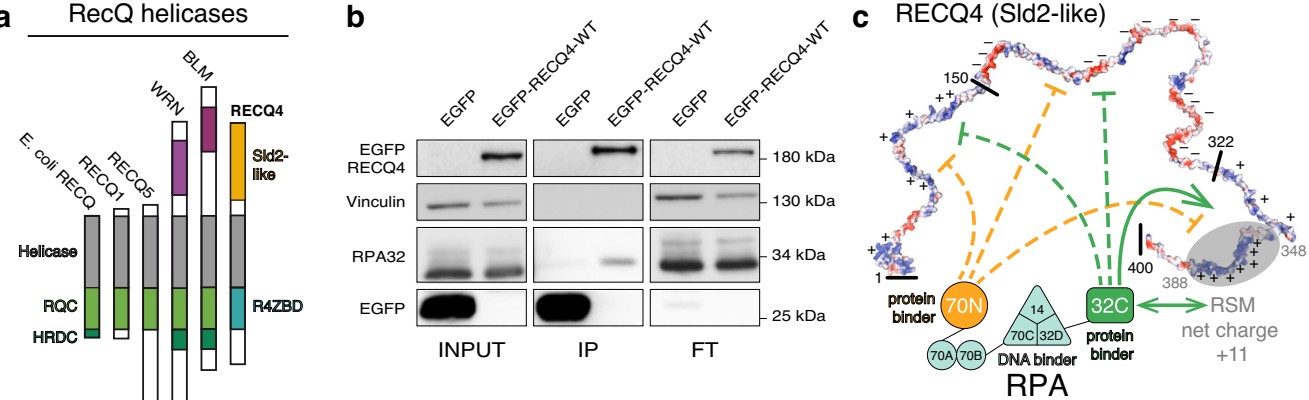

**Fig. 1 | An interaction between RECQ4 and RPA. a** Schematic representation of domain organization across the RecQ helicase family. **b** Immunoprecipitation of EGFP-RECQ4-WT or EGFP from whole cell lysates by GFP-trap beads. Samples were run on SDS-PAGE gel and immunoblotted on nitrocellulose membrane with indicated antibodies. The gel is a representative image of two independent experiments. **c** The topology of the RPA-RECQ4(Sld2-like) association was determined by NMR to detect physical interactions between segments. RPA modules are annotated as DNA or protein binders. For the RECQ4 fragment, RPA binding was further refined and located on RSM. Source data are provided as a Source Data file.

region of RECQ4 may contain the RPA-interacting site. Therefore, we performed a pull-down with the Sld2-like region of RECQ4 (aa 1–400) and RPA heterotrimer and confirmed the direct physical interaction (Supplementary Fig. 3a). The interaction was further characterized using 2D nuclear magnetic resonance (NMR) spectroscopy. The addition of unlabeled heterotrimeric RPA at a 2:1 molar ratio with $^{15}$N-enriched RECQ4 (1–400) caused several peaks to broaden beyond detection (Supplementary Fig. 3b). Such line broadening of NMR signals results from a substantial increase in the overall molecular tumbling rate associated with the formation of a ~160 kDa RECQ4-RPA complex.

Next, we used an NMR-based approach to determine the minimal regions required for the interaction. The Sld2-like region of RECQ4 was divided into three polypeptides based on the charge distribution, while on the RPA side, the well-studied protein interaction modules RPA70N and RPA32C were selected[92,93] (Fig. 1c). 2D Heteronuclear single-quantum coherence (HSQC) spectra of $^{15}$N enriched RECQ4 polypeptides were acquired in the absence and presence of a fourfold excess of each RPA domain. The fingerprint spectra indicate that the interaction occurs between the RECQ4 region spanning amino acids 322–400 and the RPA32C domain (Fig. 1c and Supplementary Fig. 3c, d). The specificity of the pairwise determination was corroborated by monitoring the chemical shift changes in RPA32C spectra induced by the addition of an excess of RECQ4$_{(322–400)}$ (Supplementary Fig. 4a).

To further delineate the region required for the interaction with RPA, we assigned the amino acid shifts of RECQ4$_{(322–400)}$ and classified them as perturbed or non-perturbed upon the addition of RPA32C (Supplementary Fig. 4a). This allowed us to map the RPA interaction

within a RECQ4-specific motif (RSM; aa 348–388) that is conserved among various RECQ4 homologues (Supplementary Fig. 4b, c). RSM is intrinsically disordered and contains a basic patch with polyelectrolyte character composed of several positively charged residues (Figs. 1c and 2e). A complete set of NMR titration experiments for both RSM and RPA32C showed that the two proteins interact in a fast exchange regime in the NMR timescale (Fig. 2a, b). The larger chemical shift perturbations (CSPs) of RSM involved mainly the cationic residues of the positive patch, whereas those of RPA32C mapped to an acidic cleft on the surface of the domain (Fig. 2c, d). The NMR atomic-level view of the interaction is in excellent agreement with the common binding mode of RPA32C that utilizes the very same acidic cleft to associate with other partners, including UNG2, XPA, SMARCAL1, and TIPIN[94–96], all of them sharing positively charged residues in their interaction motif (Fig. 2d and Supplementary Fig. 5a). Unlike the other RPA partners, RECQ4 contains two well-conserved tryptophans in the binding motif. However, an RSM double tryptophan mutant (W379A/W383A) did not affect RPA32C binding in vitro as judged by the CSP binding profile (Supplementary Fig. 5b, c).

To ascertain the importance of the basic patch for the interaction with RPA, we reversed the charge in five amino acids (R375, K376, K380, K382, R384; 5E-mutant) based on the CSP quantification (Fig. 2e). As expected, the 5E-mutant abolished the binding to RPA32C in vitro (Supplementary Fig. 6). Next, we evaluated the interaction in a cell-based experiment by introducing doxycycline-inducible N-terminal EGFP-tagged RECQ4 mutant variant into U2OS cells (EGFP-RECQ4-5E) depleted for endogenous RECQ4 and performed co-immunoprecipitation experiments. We found that RPA co-precipitation is strongly attenuated with the RECQ4 5E-mutant

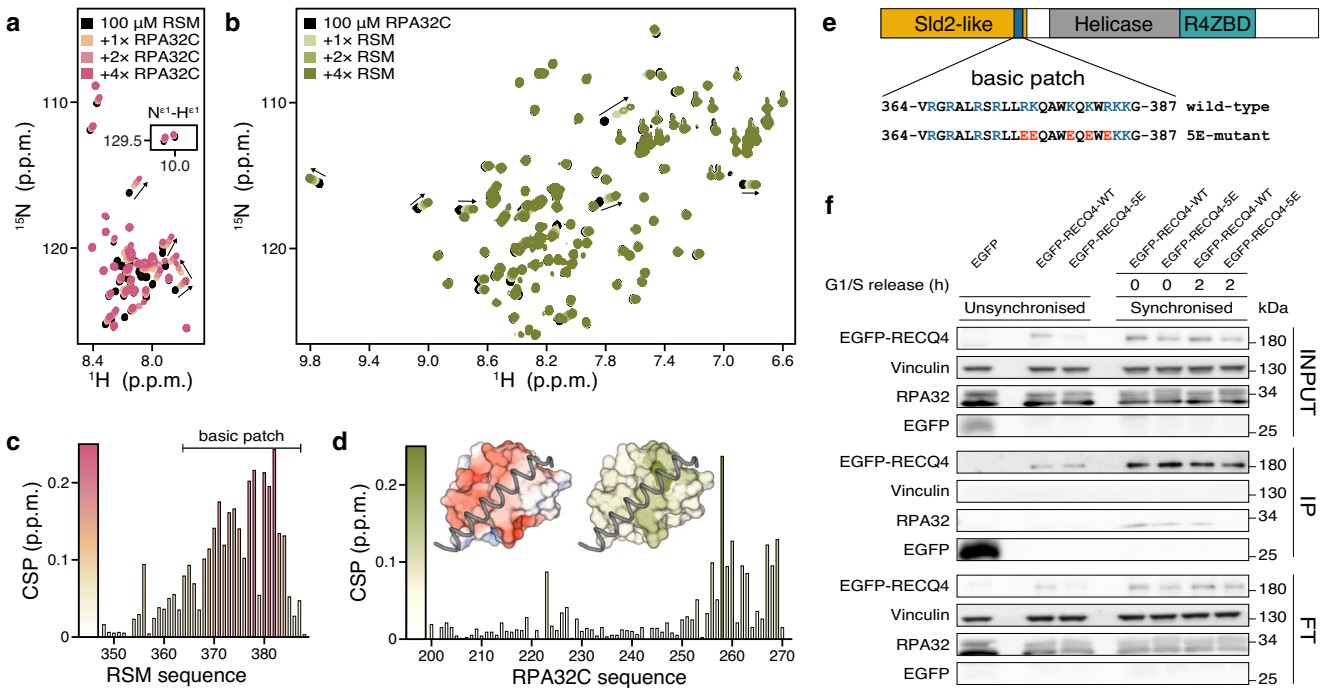

**Fig. 2 | RSM binds the acidic cleft of RPA32C and is critical for RPA-binding in vivo. a, b** Mapping the binary interaction between RSM and RPA32C by NMR titrations. $^{15}$N labeled RSM titrated with zero to fourfold molar addition of RPA32C (**a**) and the reverse (**b**). Well-resolved chemical shift perturbations (CSP) are indicated with arrows. Inset in (**a**) shows the crosspeaks of tryptophan sidechains. **c, d** Per residue amide CSP of RSM (**c**) or RPA32C (**d**) induced by fourfold excess of the unlabeled partner. (**d**, inset) Crystal structure of RPA32C in complex with the SMARCAL1 peptide (PDB: 4mqv) showing (left) RPA electrostatics and (right) mapping of the RSM-induced CSPs. **e** Schematic depiction of RECQ4 protein

showing the wild-type and 5E-mutant sequences of the basic patch. Positively charged residues are shown in blue, and charge reversal substitutions are in red. **f** Whole-cell lysates were immunoprecipitated from EGFP-RECQ4-WT, EGFP-RECQ4-5E, and EGFP unsynchronised or synchronised cells using GFP-trap beads. Bound proteins were separated on SDS-PAGE gel and immunoblotted with indicated antibodies on the nitrocellulose membrane. IP samples for RPA32 represent higher exposure of the membrane. The gel is a representative image of two independent experiments. Source data are provided as a Source Data file.

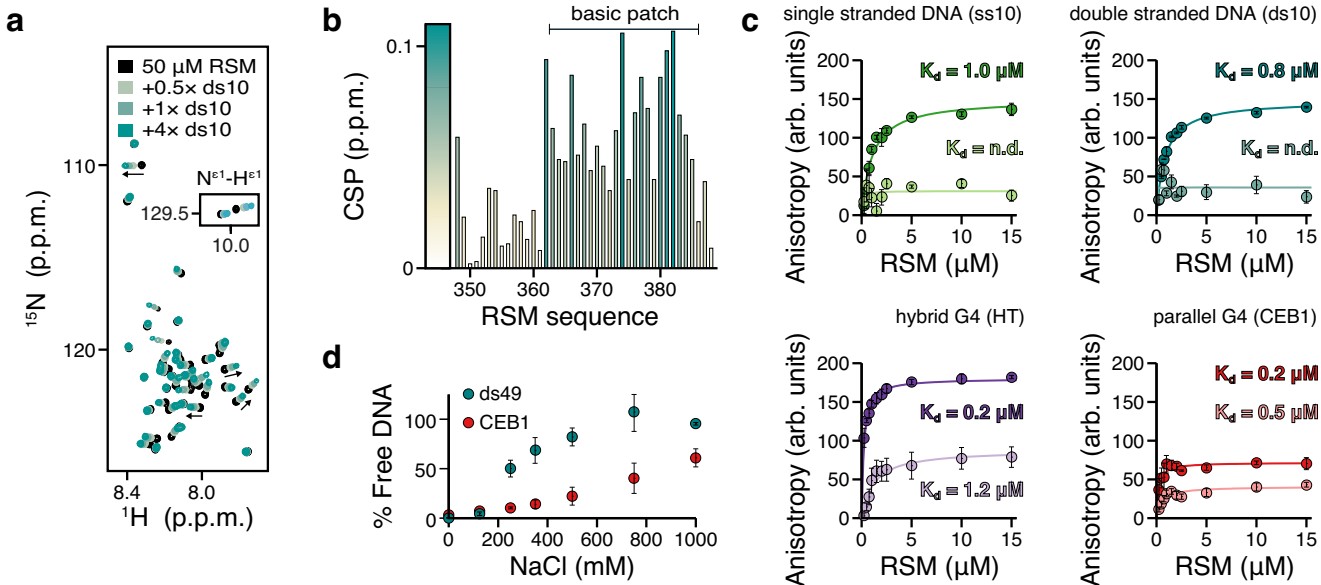

**Fig. 3 | RSM forms electrostatically driven high-affinity complexes with DNAs.**
**a** $^1$H-$^{15}$N HSQC spectra of 50 μM RSM titrated with double-stranded 10mer DNA (ds10). Some chemical shift perturbations (CSP) are indicated with arrows. Inset shows the crosspeaks of tryptophan sidechains. **b** CSPs of RSM residues induced by 4× molar addition of ds10. **c** Binding of RSM (dark color, 70 mM ionic strength; light color, 500 mM ionic strength) to ss10, ds10, HT, or CEB1 DNA monitored by FA measurements. $n = 3$ independent experiments; data are means ± s.d. **d** Effect of ionic strength on RSM binding to ds49 or CEB1 DNA quantified in EMSA experiments (see Supplementary Fig. 7f). $n = 3$ independent experiments; data are means ± s.d. Source data are provided as a Source Data file.

ex vivo, confirming that the basic patch of RSM is the principal determinant of the RPA-RECQ4 interaction (Fig. 2f).

## RSM forms electrostatically driven high-affinity complexes with various DNA structures

The Sld2-like region of human RECQ4, particularly the RECQ4$_{(322-400)}$ fragment, has been previously shown to bind various DNA substrates with a strong preference for G4s[56,57,65,98]. To elucidate which part of RECQ4$_{(322-400)}$ possesses the DNA binding activity, we inspected the chemical shift changes of this fragment when bound to a 10-bp double-stranded DNA (ds10). The NMR analysis showed that only the residues within the RSM are affected by the presence of the DNA substrate (Supplementary Fig. 7a). To support this finding, we performed an Electrophoretic Mobility Shift Assay (EMSA) of various RECQ4 peptides within the 322–400 region with a 49-bp dsDNA (ds49) substrate. The complex binding profiles observed were indistinguishable, suggesting similar affinities for all tested peptides (Supplementary Fig. 7b, c). Our data thus demonstrate that the RSM itself or its shorter version obtained as a synthetic peptide (sRSM: 358–388) is sufficient for the DNA binding properties of the RECQ4$_{(322-400)}$ fragment[56].

Next, we monitored the interaction by NMR titrations to obtain atomic-level insight into the DNA binding properties. The RSM binds ds10 DNA in a fast exchange regime on the chemical shift timescale (Fig. 3a). CSP analysis showed that DNA binding is mediated by the positively charged residues of the basic patch (Fig. 3b), highlighting the dominant role of electrostatics in the interaction. We also compared the CSPs induced by other DNA substrates. The overall pattern was the same for a splayed-arm (Y14), single-stranded (ss10), or double-stranded DNA (ds10) with minor differences in the relative magnitude of CSPs (Supplementary Fig. 8). These data point strongly to polyelectrolyte interactions dominated by non-specific electrostatic contacts between the flexible cationic RSM and the negatively charged DNA molecules. Importantly, the binding of RSM to dsDNA did not affect the integrity of the DNA helix (Supplementary Fig. 7d). Interestingly, the CSP maps of the RSM bound to DNAs or RPA largely overlapped, suggesting that the RSM utilizes the same basic patch for different functions.

To examine the role of electrostatics in DNA binding, we performed Fluorescence Anisotropy (FA) experiments with ss10, ds10, and two G4 structures, one with hybrid (HT)[99] and one with parallel (CEB1)[100] topology (Supplementary Table 1). At low ionic strength, the binding affinities were in the submicromolar range for both G4s and around one micromolar for ss10 and ds10 DNAs (Fig. 3c). EMSA experiments further confirmed the RSM preference for G4 structures. In a mixed pool of equal amounts of CEB1 G4 and ss20 DNA, RSM bound the G4 structure with greater affinity than the ssDNA substrate (Supplementary Fig. 7e). At high ionic strength, binding assessed by FA was attenuated for the G4s and not reliably detected for both ss10 and ds10 DNAs (Fig. 3c). EMSA with increasing amounts of salt further supported the electrostatic nature of the DNA interactions. However, consistent with the FA results, RSM binding to CEB1 was more resistant to ionic strength than ds49 (Fig. 3d and Supplementary Fig. 7f). This data suggests that the versatile DNA binding properties of the RSM depend on ionic strength.

## RSM conformational flexibility modulates distinct and competitive functions

Intrinsic disorder in proteins is often associated with promiscuous binding[101,102]. Indeed, our data demonstrate that RSM employs the physicochemical features of the basic patch to associate with RPA or various DNA substrates. To obtain detailed mechanistic information on RSM binding plasticity, we characterized the kinetic and thermodynamic parameters of the interactions along with structural and dynamic NMR descriptors.

A comparison of NMR titration measurements revealed that the relatively strong binding of RSM to dsDNA is associated with only small RSM chemical shift changes (CSPs not larger than 0.1 p.p.m., Fig. 3b). In contrast, the weaker binding of RSM to RPA32C is associated with much larger RSM chemical shift changes (CSPs as large as 0.25 p.p.m., Fig. 2c). These observations suggested that RSM complex formation with dsDNA or RPA32C may involve distinct binding mechanisms. To explore this, we compared the structural and dynamical traits of the RSM (Fig. 4a). Analysis of NMR secondary chemical shifts ($^1$H$^\alpha$, $^1$H$^N$, $^{15}$N, $^{13}$C′, $^{13}$C$^\alpha$, $^{13}$C$^\beta$) provides a sensitive indicator of dynamics and

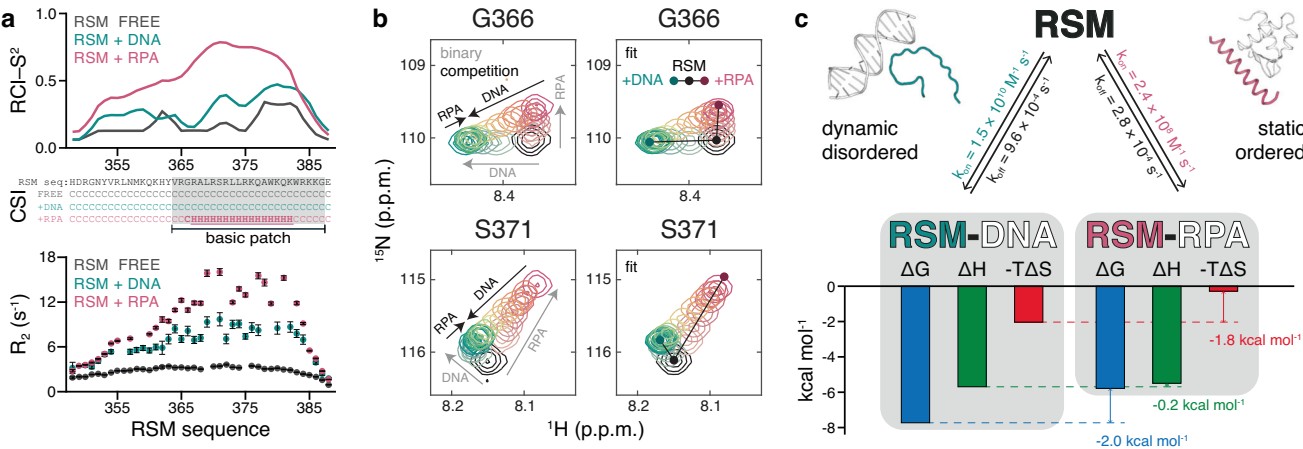

**Fig. 4 | Distinct and competitive binding modes of RSM to DNA and RPA.**
**a** Random Coil Index order parameters (RCI-S²) and secondary structure from Chemical Shift Index (CSI) (C: random coil, H: helix) calculated from secondary chemical shifts and $^{15}$N $R_2$ relaxation rates measured for free RSM (black; 50 μM RSM) and RSM in complex with DNA (cyan; 4× molar excess dsDNA) or RPA (pink; 16× molar excess RPA32C). Relaxation experiments were performed once. Error bars represent the standard error of the fitted parameters. **b** Competitive binding of DNA (4× molar excess) and RPA (16× molar excess) to 50 μM RSM, observed by 2D NMR titrations and shown for two residues of RSM. Gray arrows indicate peak trajectories upon binding of each partner, and black arrows indicate peak trajectories upon competition with the other partner (left). Fitted spectra following 2D line shape analysis of the competition binding (right). **c** Summary of kinetic, thermodynamic, and structural parameters of RSM interactions. Source data are provided as a Source Data file.

structure[103–105]. The chemical shifts of free RSM indicated a highly disordered peptide with no intrinsic propensity for secondary structure formation (Fig. 4a and Supplementary Table 2). Similarly, the RSM chemical shifts in complex with dsDNA showed that it remains relatively flexible and unstructured (Fig. 4a and Supplementary Table 3). In contrast, the RSM chemical shifts in complex with RPA32C indicated increased order within the basic patch of RSM, associated with the formation of a stable α-helix (Fig. 4a and Supplementary Table 4). This is in line with studies of other peptides which undergo a disorder-to-helix transition upon RPA32C binding[94–96] (Fig. 2d and Supplementary Fig. 5b).

We also performed $^{15}$N $R_1$, $R_2$, and {$^1$H}–$^{15}$N heteronuclear NOE (nuclear Overhauser effect) relaxation measurements to characterize the dynamic behavior of RSM (Fig. 4a and Supplementary Fig. 9a). The $^{15}$N relaxation rates of free RSM are characteristic of a disordered peptide with no elements of increased rigidity. Binding to either dsDNA or RPA32C is mainly characterized by elevated $R_2$ values for the basic patch, which could be due to changes in rotational diffusion or due to slower processes like folding. Further analysis of the relaxation rates based on spectral density mapping[106,107] (Supplementary Fig. 9b, c) demonstrated internal rigidity for the basic patch in complex with dsDNA, which increased further when in complex with RPA32C. The relaxation data agree very well with the structural information encoded by the secondary chemical shifts and the entropy changes associated with binding (see below).

These NMR observations demonstrate two distinct modes of binding mediated by the RSM. RSM forms a dynamic complex with dsDNA, where internal motions are reduced, but the peptide remains flexible and disordered. In contrast, binding to RPA32C is accompanied by helix formation to yield a static ordered complex. However, both binding mechanisms rely on the positively charged RSM epitope, indicating that RPA32C and dsDNA interactions could be competitive and mutually exclusive.

To test this scenario and shed light on the multifunctional nature of the RSM, we performed 2D NMR competition measurements of $^{15}$N-labeled RSM with dsDNA and RPA32C (Fig. 4b). First, RSM was titrated up to saturating quantities of dsDNA (4× excess) or RPA32C (16× excess). Then, each RSM complexed state was challenged with the competitor (Fig. 4b and Supplementary Fig. 10). In both competition

experiments, RSM peaks lay along linear paths between the peak positions of the RPA32C- or dsDNA-bound states, confirming that RPA32C and dsDNA compete for RSM binding. Regardless of the order of the addition, the equilibrium established at the end was in favor of dsDNA binding due to the stronger affinity (Fig. 4b). To quantify these interactions, we performed a 2D lineshape analysis on the complete set of four titrations. This procedure simulates the complete set of 2D NMR experiments using a virtual spectrometer approach to determine the best-fitting spectral parameters (resonance frequencies and line-widths) and equilibrium and rate constants for a given binding model[108,109]. To determine dissociation constants and dissociation rates for dsDNA and RPA32C, RSM residues were fitted globally across 24 titration points to a competitive binding model (Fig. 4b, c). A good fit was obtained for all RSM residues, and the fitting quality did not improve by incorporating direct exchange between RPA32C- and dsDNA-bound states into the binding model. This analysis indicates that exchange between the two complexes occurs predominantly via dissociation and re-binding rather than direct displacement and agrees with the distinct structural states of the RSM when it associates with dsDNA or RPA32C.

The results of the 2D lineshape fitting above indicate that the affinity of RSM for dsDNA ($K_d$ 6.3 ± 0.6 μM) is approximately 20-fold greater than for RPA32C ($K_d$ 117 ± 5 μM). As dissociation rates are roughly comparable, the different affinities are driven primarily by the difference in the association rates, which we find to be $1.5 ± 0.4 × 10^{10} M^{-1} s^{-1}$ and $2.4 ± 1.5 × 10^8 M^{-1} s^{-1}$ for dsDNA and RPA32C, respectively (Fig. 4c). Notably, the association rate for dsDNA exceeds even the simplest Smolochowski diffusion limit (ca. $10^9 M^{-1} s^{-1}$, neglecting orientational constraints)[80,110], indicating that formation of the complex is steered by long-range electrostatic forces. However, given that the RPA32C binding cleft is also strongly negatively charged, the reduced association rate to RPA32C by two orders of magnitude is likely associated with the energetically costly formation of the observed α-helix. Therefore, we examined the interaction with dsDNA or RPA32C by Isothermal Titration Calorimetry (ITC) and analyzed the binding thermodynamics. The dissociation constants determined by ITC are in good agreement with the ones obtained from NMR competition experiments, confirming a 25-fold stronger binding affinity of RSM for dsDNA (Supplementary Fig. 11). The associated

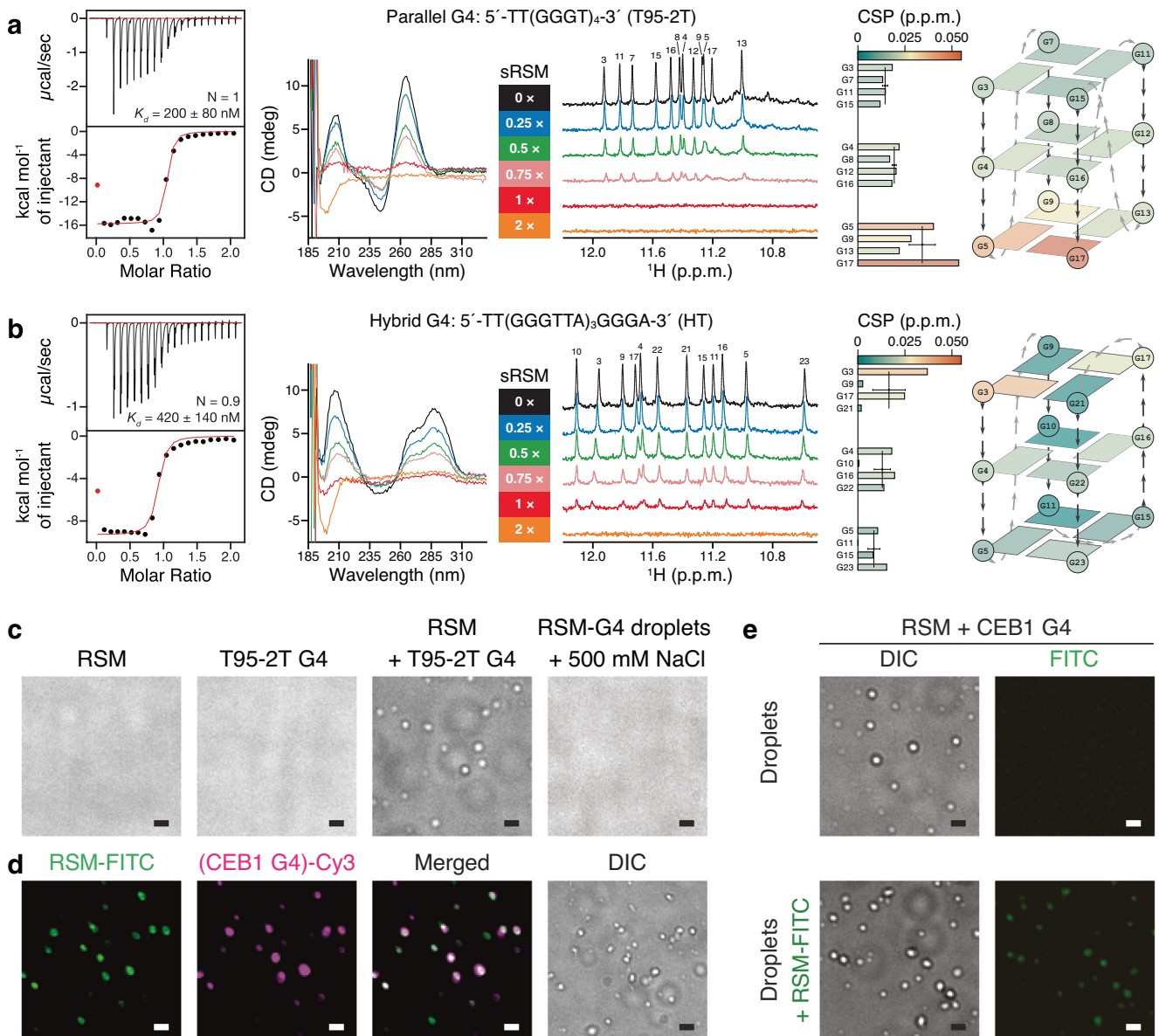

**Fig. 5 | RSM-G4 recognition and coacervation.** sRSM interaction with (**a**) parallel and (**b**) hybrid G4 was analyzed by ITC experiments (100 μM RSM), CD measurements (10 μM G4), or NMR titrations (50 μM G4). Imino chemical shift perturbations were quantified at sRSM:G4 ratio of 0.75:1 and color mapped in the schematic diagrams. G4 sequences are indicated. **c** RSM (10 μM), parallel T95-2T G4 (10 μM) or their mixture (10 μM each) were analyzed for droplet formation using DIC microscopy. The addition of 500 mM NaCl dissolves the droplets. Images are

representative of three independent experiments. **d** Droplet formation and colocalization of 5 mol % FITC-labeled RSM (10 μM) and 5 mol % Cy3-labeled CEB1 G4 (10 μM). Images are representative of three independent experiments. **e** Droplet formation (upper panels) and colocalization (lower panels) of 5 mol% FITC-labeled RSM with preformed droplets. Images are representative of two independent experiments. In all images scale bar = 1 μm. Source data are provided as a Source Data file.

thermodynamic components (ΔG, ΔH, and ΔS) highlighted that the enthalpic contribution in both interactions is very similar, whereas substantial differences were observed in the entropic contribution (Fig. 4c). The large entropy difference cannot be attributed to hydrophobic interactions because the binding of both complexes involves the same positive patch of the RSM (Fig. 4b). Instead, the conformational properties of the RSM likely account for this phenomenon[111,112]. The dynamic nature of the RSM-dsDNA complex benefits from conformational flexibility, due to entropically favored interactions between any pairs of charged residues. In contrast, the formation of a static complex with RPA32C involves specific interactions and RSM folding into a well-defined helical configuration, which minimizes any entropic contribution. We conclude that the RSM conformational entropy modulates the kinetics and thermodynamics of binding to perform mutually exclusive functions.

## Binding and coacervation between RSM and G-quadruplexes

Since RecQ family members (bacterial RECQ, WRN, BLM) are among the best-characterized helicases that unwind G4 motifs[113], we investigated the G4 binding properties of RSM in detail to understand RECQ4 roles in nucleic acid metabolism. For this, we used the T95-2T[114] and HT[99] G4s that fold into stable parallel and hybrid G4 topologies, respectively (Supplementary Fig. 12). The binding isotherms measured by ITC revealed that RSM binds to both quadruplexes with stoichiometry close to 1:1 and up to 8-fold higher affinity compared to dsDNA (dissociation constant of 200 ± 80 nM for the parallel and 420 ± 140 nM for the hybrid G4) (Fig. 5a, b and Supplementary Fig. 11). We also investigated RSM binding to G-quadruplexes by NMR. The RSM amide peaks in the HSQC spectra showed a dose-dependent reduction in their intensities upon the addition of G4s and completely disappeared at 1:1 stoichiometry (Supplementary Fig. 13a). Peak

intensities of positively charged residues were more reduced compared to the general trend (Supplementary Fig. 13b), pointing to the importance of the basic patch in G4 association. However, the fact that RSM peaks synchronously lose intensity (Supplementary Fig. 13a, b) is indicative of the possible formation of high-order complexes.

Indeed, the RECQ4 N-terminus is highly unstructured, with several regions predicted to undergo liquid-liquid phase separation (LLPS)[115]. Since the RSM shows the highest LLPS propensity (Supplementary Fig. 14a), we set out to analyze our samples for the formation of condensates by optical microscopy. While a solution of RSM or G4 alone was free of any visible particles, mixing RSM with G4 led to a rapid formation of droplets (Fig. 5c). Using fluorescently labeled components, we confirmed the presence of both RSM and G4 in the droplets (Fig. 5d). Formation of droplets was accompanied by sample turbidity that increased in a concentration-dependent manner (Supplementary Fig. 14b). We observed droplet formation with two types of parallel G4 (T95-2T and CEB1), hybrid G4 (HT), and ssDNA of varying length, with the notion that the shortest ssDNA showed very few droplets, but not with dsDNA or RPA32C (Supplementary Fig. 14d). This data indicate that RSM is prone to phase separation when mixed with oppositely charged polyelectrolytes, a process called complex coacervation that strongly depends on electrostatic interactions[116]. Indeed, RSM-DNA droplets are sensitive to salt concentration and dissolve above a critical salt concentration confirming the polyelectrolyte behavior of phase separation (Fig. 5c and Supplementary Fig. 14c). Likewise, the 5E-mutant that effectively neutralized the RSM charge did not form droplets when mixed with G4s. However, residual binding was still detectable by NMR between RSM-5E mutant and parallel G4 but not between RSM-5E mutant and hybrid G4 (Supplementary Fig. 14e, f). On the other hand, the RSM double tryptophan mutant was prone to forming droplets with both G4s, suggesting that the two tryptophan residues within the RSM are not critical for coacervation (Supplementary Fig. 14e, f).

To better understand the mechanism of RSM-G4 coacervation, we monitored the G4 proton resonances by NMR. The two selected G4 structures (T95-2T and HT) produced well-resolved imino spectra matching the assignments from the original reports and confirming G4 integrity and topology[99,114]. We performed NMR titrations by gradually adding RSM (aa 348–388) or sRSM (aa 358–388) to both G4s. The intensity of both imino and aromatic G4 peaks decreased proportionally with the dose of corresponding RECQ4 peptide to reach the noise level close to 1:1 stoichiometry (Fig. 5a, b and Supplementary Fig. 15). The progressive signal loss of G4 proton resonances reflects the droplet formation. Concurrently with the decrease of signal intensities, chemical shift perturbations were also observed, representing the first encounter between RSM and the G4 in the dilute solution phase on the path towards coacervation. Structural mapping of chemical shift perturbations in the parallel G4 suggests that the RSM engages first the 3′-end G-tetrad. In contrast, for the hybrid G4, the largest chemical shift perturbations are observed at the 5′-end G-tetrad, indicating that initial binding is sensitive to the G4 structure (Fig. 5a, b).

To further investigate the interaction between RSM and G4s, we performed circular dichroism (CD) measurements. Each G4 displays a unique CD spectral signature reflecting mainly G-quartet stacking, strand segment orientation, and loop arrangements[117]. Importantly, sRSM (aa 358–388) did not interfere with the G4 CD spectral features (Supplementary Fig. 16a). The G4-specific CD signal (at 265 nm for the parallel G4 and 286 nm for the hybrid G4) decreased in a dose-dependent fashion upon addition of synthetic RSM and disappeared at an equimolar ratio for both quadruplexes (Fig. 5a, b and Supplementary Fig. 16b). Since the CD observations correlated with the formation of droplets, we compared the CD spectra of samples in which the phase-separated sRSM-G4 were removed by centrifugation prior to measurement with those of samples containing the sRSM-G4 droplets.

Interestingly, the effect on the magnitude of the CD spectrum was very similar regardless of the presence of droplets in the sample and proportional to the G4 amount remaining in the dilute phase collected after centrifugation (Supplementary Fig. 16c). It has been shown that the packing mode of dsDNA molecules in dispersion particles can change their optical properties[118]. Although it is unclear why RSM/sRSM-G4 droplets become optically inactive while other G4 coacervates do not[119], this unexpected finding presented the means to monitor the G4 equilibrium between dilute and condensed phases.

To gain a better understanding of the coacervation process between parallel G4 (T95-2T) and sRSM, a modern kinetic analysis was conducted, combining equilibrium titration data (Fig. 6a–c) with transient stopped-flow kinetics (Fig. 6d, e) in a global numerical model. Initially, a conventional analytical fitting was performed to demonstrate: (i) the presence of three distinct transitions between the dilute and condensed phases, (ii) a 1:1 stoichiometry, and (iii) a "cooperative" nature of sRSM-DNA assembly (Supplementary Fig. 17 and Supplementary Note). Subsequently, this information and the initial estimates of kinetic constants were used to develop a rigorous numerical model. The complex dataset, comprising two independent replicates for each experiment (Supplementary Fig. 18), was simultaneously fitted using numerical integration of the rate equations derived from the proposed model (Fig. 6f). The majority of the data were fitted in their raw form, except for the CD and NMR titrations which required additional processing to account for the concentration dependencies on the observed signals. Singular Value Decomposition (SVD) analysis of CD spectra concentration dependence demonstrated high similarity to the data acquired through a simple reading at two wavelengths (198 and 265 nm), indicating a straightforward relationship between the concentrations of sRSM and G4 and the resulting CD signal. Consequently, the specific-wavelength CD readings were utilized during the fit (Fig. 6a). On the contrary, the SVD analysis of NMR spectra provided valuable insights into the complex interplay between the reaction species and their influence on both the intensity and the chemical shift perturbation observed in the NMR titrations (Fig. 6b). Therefore, the complex NMR data were represented in the global fit through the SVD amplitude vectors.

Fitting the multiple data globally enabled the derivation of a unique set of rate constants for the individual steps of the proposed model for sRSM-G4 coacervation (Fig. 6f). The initial binding between sRSM and G4 occurred rapidly and reached equilibrium within 100 ms (Fig. 6d). The initial complex was relatively weak, with a dissociation constant ($K_{d,1}$) of $3.1 \pm 0.2\,\mu M$. The formation of the initial sRSM-G4 complex was slightly unfavorable, resulting in a positive free energy change ($\Delta G_{0,1}$) of 2.83 kJ mol$^{-1}$ ($\Delta G$ values are reported for a reference state of 1 μM). The second step of the coacervation process led to the formation of a stable nucleus ($n = 2$, the minimum value satisfying the model) with a dissociation constant ($K_{d,2}$) of $95 \pm 7$ nM. The free energy change ($\Delta G_{0,2}$) associated with nucleation is $-5.71$ kJ mol$^{-1}$, strengthening the sRSM-G4 complex stability. The stable nucleus served as the foundation for subsequent growth ($n \geq 3$), which ultimately led to phase separation driven by a change in free energy ($\Delta G_{0,3}$) of $-3.88$ kJ mol$^{-1}$ (Fig. 6g). The comprehensive kinetic analysis revealed that the coacervation of G4 and sRSM follows a rapid nucleation-growth mechanism. The simulation of the concentration of individual states (Fig. 6h) provided a visual representation of the dynamic equilibrium between the dilute and condensed phases under different conditions. The global numerical analysis provided a χ²/DoF value of 1.12, indicating a good fit where an average global variance value is comparable to the average internal variance of the data. To further validate the accuracy of the model, we conducted spin-down assay experiments probing for the dilute species under varying concentrations. The experimental and simulated data (Fig. 6i and Supplementary Fig. 19) confirmed the consistency of the predictions and the reliability of the model in accurately capturing the dynamics of the system.

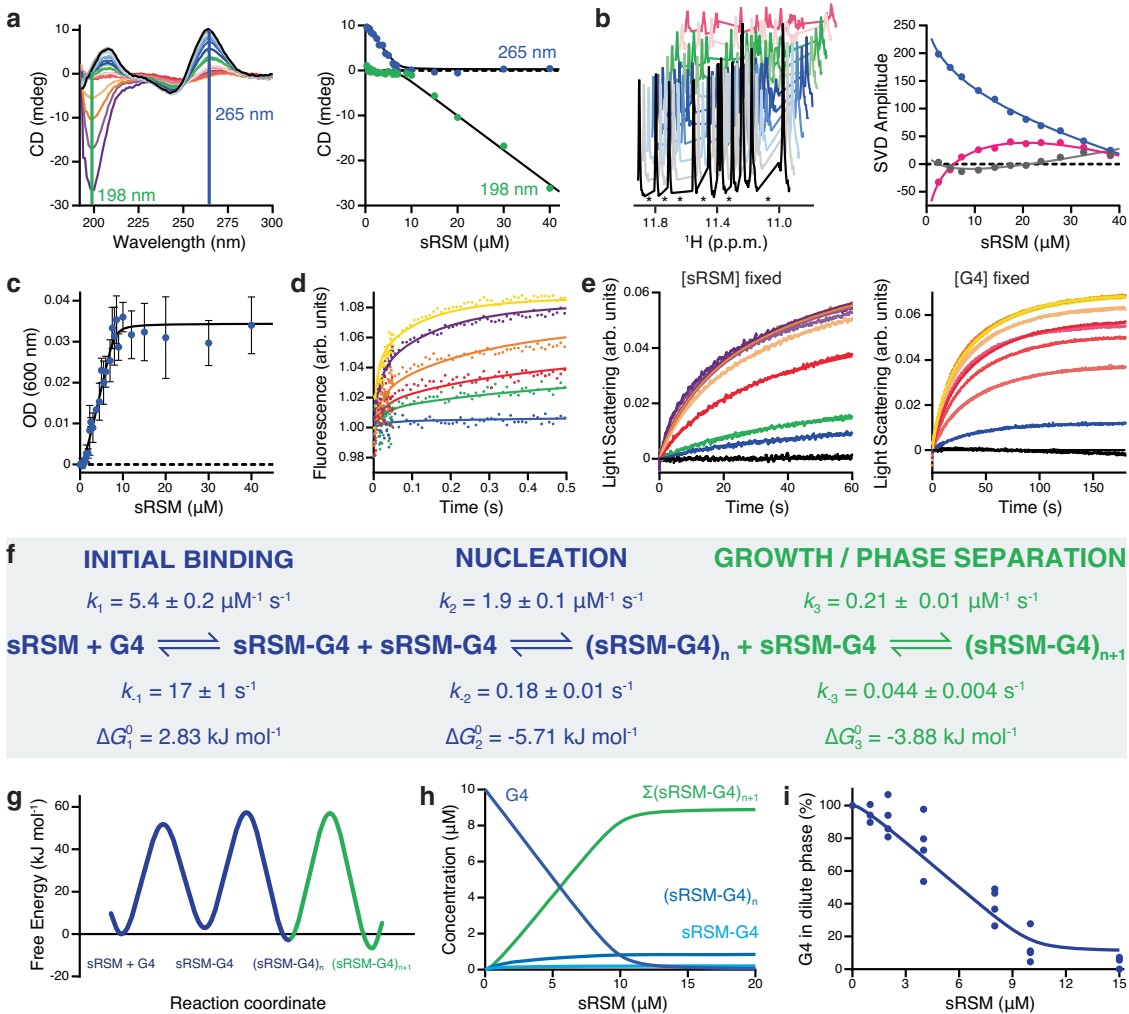

**Fig. 6 | Kinetic model of sRSM-G4 coacervation. a** CD spectra of 10 μM G4 titrated with 0–40 μM sRSM. The CD signal readings (right) at two specific wavelengths, 198 nm and 265 nm, were utilized in the global numerical analysis. **b** SVD amplitudes (right) calculated from the dependence of 50 μM G4 imino NMR spectra (left) on increasing sRSM concentration. Asterisks mark removed baseline signal regions to improve accuracy in SVD analysis. **c** The optical density at 600 nm monitored during the titration of 10 μM G4 with 0–40 μM sRSM. n = 3 independent experiments; data are means ± s.d. **d** The initial phase of the reaction examined by stopped-flow fluorescence upon mixing 2.5 μM G4 with 0–20 μM sRSM. **e** Kinetics of condensation process monitored by light scattering (at 295 nm) upon rapid mixing of 2.5 μM sRSM with 0–20 μM G4 (left) or 2.5 μM G4 with 0–20 μM sRSM (right). Each trace (in **d** and **e**) represents the average of 3 or 4 measurements. Solid lines in graphs from (**a**) to (**e**) represent the best global fit to the kinetic data. **f** The

minimal kinetic model describing the sRSM-G4 coacervation includes three main steps: (i) initial binding, (ii) formation of stable nuclei and (iii) growth leading to phase separation. The rate constants for the individual steps were derived from global numerical fit. **g** Free energy profile of sRSM-G4 assembly and the reported free energy values in (**f**) were calculated using the Eyring equation (298 K; 1 μM RSM; 1 μM sRSM-G4). **h** The simulation represents the dynamic equilibrium of individual species for 10 μM G4 and sRSM concentration ranging from 0 to 20 μM. **i** The concentration of soluble species monitored using spin-down assay after incubation of 10 μM G4 with varying concentration of sRSM. Solid line represents the simulated data of G4 in dilute phase (sum of G4, sRSM-G4 and (sRSM-G4)$_2$ species), obtained from a global numerical model; data points from 4 experiments. Source data are provided as a Source Data file.

To demonstrate the dynamic nature of RSM-G4 coacervation, we performed further NMR experiments to perturb the final equilibrium. First, we introduced an excess of unlabeled sRSM to the stoichiometric mixture of [15]N RSM in complex with G4 that had already undergone phase-separation. This resulted in redistribution of the previously unobservable phase-separated [15]N RSM molecules between the dilute and condensed phases, resulting in the recovery of the NMR signal from the free [15]N RSM population in the dilute phase (Supplementary Fig. 13a). Consistently, FITC-labeled RSM was able to enter preformed RSM-G4 droplets (Fig. 5e). In a complementary experiment, we added an extra equivalent of G4 to a stoichiometric sRSM-G4 mixture that had formed droplets and showed no proton signals (Supplementary Fig. 13c). The resulting imino signals of excess G4 showed the sRSM induced perturbations, indicating a weighted average between the free and sRSM-bound populations in the dilute phase (Supplementary

Fig. 13c and Fig. 5a). Additionally, the extra G4 resulted in the reappearance of sRSM methyl signals in the NMR spectrum, because of increased sRSM fraction (free or in complex with G4) in the dilute phase (Supplementary Fig. 13c). These data demonstrate further the exchange of sRSM and G4 between the solution and droplets.

To assess the G4 structural state across the phase equilibrium, we took advantage of Thioflavin T, a fluorescent G4-specific probe. Thioflavin T fluorescence is markedly increased upon binding to various G4s compared to ssDNA[120,121]. Indeed, we could observe a substantial increase in Thioflavin T fluorescence when incubated with T95-2T G4 but not with ssDNA (Fig. 7a, b). Intriguingly, a similar increase in fluorescence was observed upon induction of coacervation by adding RSM (Fig. 7a, b). These findings were further substantiated by visualizing Thioflavin T directly in the droplets by fluorescence microscopy (Fig. 7c, d). Although we can not exclude the possibility that ThT signal

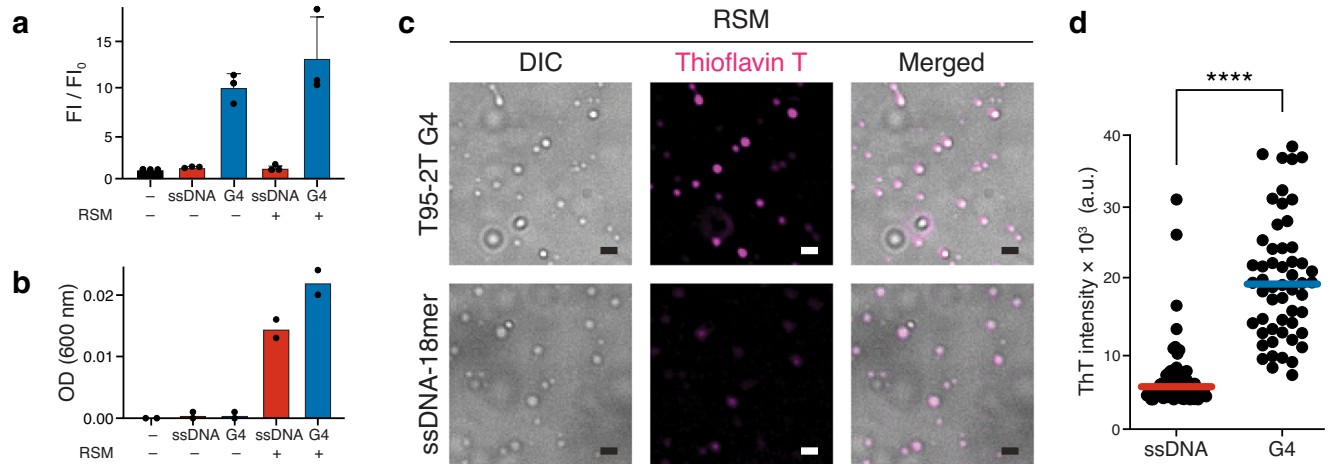

**Fig. 7 | Analysis of G4 structure inside the droplets using Thioflavin T probe.**
**a** Thioflavin T (ThT, 500 nM) was mixed with T95-2T G4 (10 μM) or ssDNA−18mer (10 μM) in the absence or presence of RSM (10 μM) and the bulk fluorescence intensity was quantified. $n = 3$ independent measurements; data are means ± s.d.
**b** Turbidity measurements of samples used for ThT fluorescence quantification in (**a**). Turbidity was measured as light absorbance at 600 nm. $n = 2$ independent measurements. **c** Phase separation microscopy of RSM (10 μM) mixed with T95-2T

G4 or ssDNA−18mer (10 μM). ThT was added to a final concentration of 500 nM and the ThT fluorescence signal in the droplets was analyzed by fluorescent micro-scopy. In all images scale bar corresponds to 1 μm. **d** Manual quantification of ThT intensity in individual droplets from images in (**c**), $n = 54$ droplets for G4 and $n = 68$ droplets for ssDNA; ****$p < 0.0001$ (two-tailed Mann−Whitney test). Source data are provided as a Source Data file.

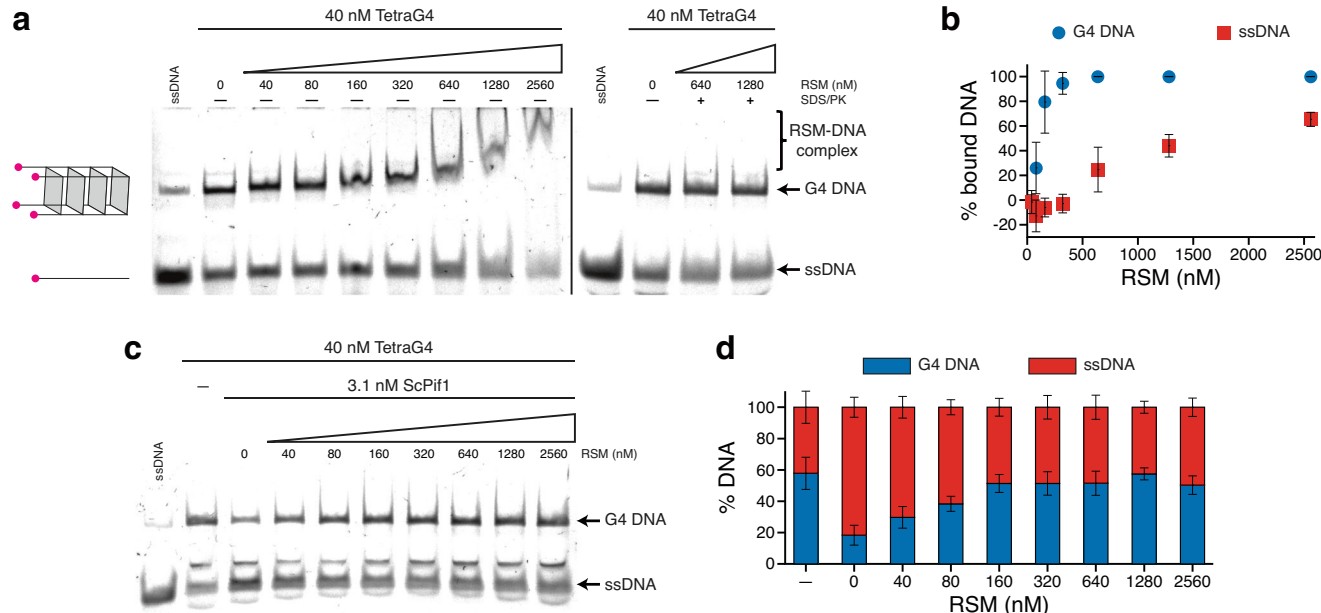

**Fig. 8 | RSM binding to a tetramolecular G4 hinders G4 processing by helicases.**
**a** Binding of increasing RSM concentrations to 40 nM fluorescently labeled tetra-molecular G4 (TetraG4) and ssDNA. Formation of RSM-G4 complex is reversed by addition of SDS/proteinase K. Schematic representation of ssDNA and folded G4 is shown on the left. **b** Quantification of gel image shown in (**a**); $n = 3$ independent

experiments; data are means ± s.d. **c** ScPif1 helicase (3.1 nM) effectively unwinds tetramolecular G4 (40 nM), but its helicase activity is inhibited by pre-incubation of TetraG4 with RSM. **d** Quantification of gel image shown in (**c**), $n = 4$ independent experiments; data are means ± s.d. Source data are provided as a Source Data file.

may partially result from RSM amyloid structures formed in the droplets[122], our data suggest that G4 retains its molecular structure in the coacervate phase.

Searching for an RSM function related to G4 binding, we assem-bled a tetramolecular G4 structure previously used to monitor helicase unwinding activity[123]. CD measurements and binding assay with BG4 antibody verified tetramolecular G4 formation (Supplementary Fig. 20a, b). The addition of RSM did not result in G4 destabilization in contrast to unwinding by yeast Pif1 helicase (Supplementary Fig. 20c, d). Yet, EMSA verified the ability of RSM to bind tetramolecular G4

(Fig. 8a, b). We also tested the effect of RSM binding to G4 on Pif1 activity and observed robust inhibition of the helicase activity (Fig. 8c, d), indicating that the RSM-G4 complex represents a barrier for G4 unwinding. Similar inhibition of Pif1 was detected using RECQ4$_{(1–400)}$ (Supplementary Fig. 21a), confirming the RSM blocking activity in the context of the Sld2-like region. This inhibition was not limited to Pif1 helicase, as RSM also blocks G4 unwinding by another helicase, FANCJ (Supplementary Fig. 21b). Taken together, our data show that RSM exhibits high affinity for G4, which hinders G4 processing. However, the RSM-mediated inhibition of G4 unwinding is not solely a

consequence of associative phase separation, as we did not detect droplet formation by light microscopy under the conditions used in these experiments. In addition, a fusion of RECQ4$_{(322-400)}$ to APEX2 tag prevented visible phase separation in higher G4 concentrations (10 μM) but did not affect G4 binding and inhibition of G4 unwinding in lower G4 concentrations (40 nM) (Supplementary Fig. 22). These results highlight the specific interaction between RSM and G4, leading to the formation of a stable complex that interferes with the activity of helicases involved in G4 resolution.

## Discussion

We have identified a positively charged region in human RECQ4 capable of forming polyelectrolyte complexes with oppositely charged nucleic acids. Interestingly, the DNA molecule profoundly affects the coacervate phase behavior of RSM-DNA complexes. G-quadruplexes exhibit a strong tendency to coacervate with RSM, whereas RSM-dsDNA complexes were not incorporated into droplets. RSM coacervation with ssDNA seems to depend on DNA length, possibly reflecting the total charge of the DNA molecule. However, the different macroscopic behaviors observed are not total-charge dependent because ds10, ss18, and G4 molecules (T95-2T and HT) have a similar net charge of around −20. Recent studies suggest that the complex coacervation of linear polypeptides is governed by charge patterning and/or charge density[124,125]. In the case of nucleic acids, molecular geometry affects the surface-charge patterning. For example, the disordered region of histone linker H1 promoted coacervation by increasing the charge density of the DNA via structuring[119]. These studies correlated rigidity and charge density of ssDNA, dsDNA, and G4 molecules having the same number of charges and ability to form droplets with H1[119]. The behavior of coacervates formed between a polylysine peptide and ssDNA or dsDNA demonstrated a similar effect depending on the stiffness and charge density of the DNA molecule[126]. Our observations on RSM-DNA coacervation differ markedly in that complexation with dsDNA did not result in droplet formation. This suggests that the electrostatic interactions that drive complex coacervation are not only influenced by the DNA charge distribution but also by the sequence (and charge) patterning of the disordered cationic peptide[85].

IDRs are characterized by high conformational entropy used to regulate their functions and interactions[127]. Our analysis demonstrates that RSM can switch between conformational states and perform diverse functions. On one hand, RSM remains highly flexible in polyelectrolyte interactions with nucleic acids. The disordered state of RSM in these complexes minimizes the entropic penalty of binding, resulting in relatively strong but non-specific interactions. On the other hand, RSM forms a stable helix when it interacts with RPA protein. The induced folding comes at an entropic cost, resulting in an interaction characterized by high specificity but low affinity. NMR competition experiments show that the two binding states are mutually exclusive. Although polyelectrolyte interactions with nucleic acids are favored over RPA32C binding in vitro, the spatiotemporal abundance of RPA molecules in the cell could trigger the transition between the disordered and ordered states of RSM. In other words, the conformational entropy of RSM can be modulated to rapidly regulate or coordinate sequential functions in cellular processes involving the RECQ4 helicase. However, we cannot exclude the possibility that the RSM-RPA32C interaction may be stabilized by other regions in the context of the RPA heterotrimer.

It is now recognized that protein IDRs are mechanistically involved in a wide range of cellular processes[128]. Our study extends this knowledge on the functions of IDRs, as it describes G4 binding within an IDR that hinders G4 processing by other helicases. RSM charge and inherent flexibility allow it to engage in strong electrostatic interactions with oppositely charged polyelectrolytes, such as G4 structures. Once in the polyelectrolyte complex, the G4 structure becomes inaccessible to processing by other helicases. Access to G4 is blocked due to RSM-G4 complex formation in solution and not because of subsequent coacervation observed in vitro. Based on our data, we speculate that the engagement of the disordered RSM with G4 represents a barrier for processing enzymes implying a possible signaling function of RECQ4 during temporal activation of replication origins[52,54,129]. Such a function may also provide mechanistic insights into the key role of RECQ4 in telomere[130,131] and mitochondrial[132,133] DNA maintenance or repair of DNA double-strand breaks[134–136] due to their direct G4 relevance[13,137].

Interestingly, these RSM properties seem to be evolutionary conserved since the *C. elegans* SLD2 protein, which shares the RSM physicochemical features, also forms droplets with G4 structures in vitro (Supplementary Fig. 23). However, a fusion of RECQ4$_{(322-400)}$ to APEX2, while not affecting G4 binding, eliminated charge-driven phase separation. This implies that the process may be regulated by structured domains within full-length RECQ4. With these considerations in mind, the great challenge ahead is to establish the localization of RECQ4-G4 complexes in phase-separated structures in the cell and the functional consequences[138].

In summary, our study uncovered an IDR in human RECQ4 with polyelectrolyte character. The physicochemical properties of the IDR account for the conformational plasticity in diverse functions and coacervation with oppositely charged polyelectrolytes such as G4 structures. Our findings offer interesting perspectives in understanding RECQ4 function in the cell linked to the partitioning of proteins into condensates.

## Methods

### Cell Culture

U2OS Flp-In T-Rex cell line (obtained from MRC PPU Reagents, University of Dundee in 2018) was maintained in DMEM media (LM-D1108/500; Biosera) with 10% tetracycline-free FBS (FB-1001T/500; Biosera) supplemented with Glutamine (XC-T1715/100; Biosera) and penicillin-streptomycin (XC-A412/100; Biosera) along with 100 μg/mL hygromycin (10687010; Invitrogen).

### Stable cell line generation and cloning

To create EGFP-RECQ4-WT plasmid, RECQ4 DNA having EGFP and HA tag at N- and C-terminus, respectively, were cloned in the EcoRV sites of pAIO plasmid[139] (vector map is shown in the Source Data of the article) with a doxycycline-inducible system containing the modified promoter of medium strength to match the endogenous expression level of RECQ4. The plasmid also contains the shRNA target sequence (5′-TAGGAAGAGCCTCATCTAAG-3′) cloned between BglII and HindIII sites. Codon optimization was done against shRNA and siRNA in the RECQ4 CDS sequence. siRNA-RECQ4 S17991 (4392420, Thermo Fisher Scientific) was used to ensure maximum RECQ4 depletion with lipofectamine RNAiMAX (13778075, Invitrogen) for 48 h. To generate RECQ4 mutant (5E) or EGFP control cells, the corresponding mutations or stop codon respectively was introduced in the EGFP-RECQ4-WT plasmid by site-directed mutagenesis. Finally, stable cell lines were generated, and individual plasmids (500 ng) were transfected into U2OS Flp-In T-Rex cells (MRC PPU Reagents; University of Dundee) using Lipofectamine™ 3000 Transfection Reagent (L3000001; Invitrogen) along with Flp recombinase expression vector (1.5 μg) according to manufacturer's protocol and were selected with 100 μg/mL hygromycin. Expression of RECQ4 constructs was induced by 1 μg/mL Doxycycline (D9891; Sigma-Aldrich) for 48 h.

### Co-immunoprecipitation

Immunoprecipitation was performed using GFP-Magnetic Agarose beads from a GFP-Trap Magnetic Agarose kit (gtmak-20; Chromotek) using 500 μg of whole cell lysate. Briefly, the cells were lysed in lysis buffer (provided by the IP kit), and the lysate was incubated with beads

for 1 h on a rotator at 4 °C, and 10% of lysate was kept as input. After incubation, the beads were separated on a magnetic separator, and 10% of the flow through was retained for analysis. The beads were washed three times with washing buffer, and the bound proteins were eluted by 2× SDS Laemmli buffer, heated at 100 °C for 5 min. All fractions were analyzed by SDS-PAGE, followed by western blotting with corresponding antibodies.

## Synchronization of cells

Cells were synchronized by a double thymidine block. Cells were seeded, and thymidine (T1895; Sigma-Aldrich) was added to a final concentration of 2 mM for 16 h. After that, cells were released in fresh media without any thymidine for 6–8 h, followed by the new addition of thymidine for another 16 h and finally released in fresh media. Cells were then harvested according to the indicated time.

## Western Blot

All samples were prepared in 2× SDS Laemmli buffer and separated on 10% SDS-PAGE at 100 V, followed by transfer of proteins to nitrocellulose membrane using the semi-dry Trans-blot turbo Transfer system (1704150; Biorad). After transfer, membranes were blocked in 5% milk/TBST for 1 h at room temperature and then incubated at 4 °C on a rocker overnight with the corresponding primary antibodies (Anti-RPA32, 1:700 dilution, A300-244A, Bethyl Laboratories; Anti-Vinculin, 1:5000 dilution, ab129002, Abcam; Anti-GFP, 1:2500 dilution, ab290, Abcam). The next day, the membranes were washed with TBST and incubated with the corresponding secondary antibodies (Anti-Rabbit IgG, 1:10,000 dilution, A6154, Sigma-Aldrich; Anti-Mouse IgG, 1:15,000, A0168, Sigma-Aldrich) for 1 h at room temperature. Finally, the blots were developed by the Immobilon Western Chemiluminescent HRP Substrate (WBKLS0500; MERCK Millipore), and images were acquired using the Luminescent Image Analyser (ImageQuant™ LAS 4000; Fujifilm).

## DNA constructs and mutagenesis

Sequences of *H. sapiens* $RECQ4_{(1-150)}$, $RECQ4_{(150-315)}$, $RECQ4_{(322-400)}$, $RECQ4_{(348-388)}$ (RSM), and $RECQ4_{(1-400)}$ (UniProt code: O94761), *C. elegans* SLD2 protein (UniProt code: O44761), *H. sapiens* $RPA32_{(172-270)}$ (UniProt code: P15927), and *H. sapiens* $RPA70_{(1-120)}$ (UniProt code: P27694) were cloned in pET-M11 plasmid between NcoI and BamHI sites (vector map is shown in the Source Data of the article) to generate $His_6$ tag– TEV cleavage site fusions at the N-terminus of corresponding proteins.

DNA fragments were PCR amplified using Phusion high fidelity DNA polymerase master mix (M0530; NEB) and the primers listed in Supplementary Table 5, excised by restriction endonucleases (NcoI, R3193; BspHI, R0517; BamHI, R0136; NEB), and ligated using T4 DNA Ligase (M0202, NEB) to pET vector already treated with Antarctic Phosphatase (M0289; NEB). DNA sequences were confirmed by sequencing.

The RSM W379A/W383A and RSM 5E-mutants were generated by PCR-based site-directed mutagenesis (primers listed in Supplementary Table 5) using Pfu Turbo DNA polymerase (600250; Stratagene) and the $RECQ4_{(348-388)}$ expression vector as template. DNA sequences were confirmed by sequencing.

$FLAG-APEX2-RECQ4_{(322-400)}$ fragment was generated in an already existing plasmid pET-11d-FLAG-APEX2-RAD51. $RECQ4_{(322-400)}$, flanked by NcoI and SalI sites, was ligated into a vector digested by NcoI and XhoI, inserting a stop codon after RECQ4.

## Protein preparation

Individual pET-M11 expression vectors (RECQ4, SLD2, and RPA) were transformed into *Escherichia coli* BL21(DE3) expression cells (C2527; NEB). Cells were grown at 37 °C in the presence of kanamycin. For $^{15}$N- and $^{13}$C-labeled samples, cells were grown in a minimal medium

supplemented with $^{15}NH_4Cl$ (0.5 g/L) and $^{13}C_6$ glucose (2 g/L) as the sole nitrogen and carbon sources, respectively. Protein synthesis was induced by the addition of 0.5 mM of isopropyl-1-thio-d-galactopyranoside (A4773; Alchimica) at $OD_{600}$ ~ 0.8, and cells were harvested after 5–6 h. Cells were resuspended in lysis buffer (25 mM Tris-HCl [pH 8], 400 mM NaCl, 20 mM imidazole, 2 mM beta-mercaptoethanol (BME), 0.1% (v/v) Triton X-100), lysed by sonication and centrifuged at $35,000 \times g$.

SLD2 and most RECQ4 proteins were purified in three steps. Initially, the lysate (denatured in 6 M urea) was applied to a HiTrap IMAC HP column charged with $CoCl_2$ (GE Healthcare) and eluted in buffer without urea supplemented with 500 mM imidazole. The protein-containing fractions were incubated at 4 °C with 0.5 mg TEV protease (produced in-house) per 1 L culture for 3 h at 25 °C to cleave the tag, followed by dialysis overnight at 4 °C against 25 mM Tris-HCl [pH 8], 100 mM NaCl, and 2 mM BME. The dialyzed protein sample was passed through a Mono S 4.6/100 PE column (GE HealthCare) and gradient eluted with 10 column volumes of buffer supplemented with 2 M NaCl. The protein-containing fractions were combined, concentrated and applied on HiLoad 10/300 Superdex Increase 75 GL column (GE HealthCare).

For the purification of $RECQ4_{(150-315)}$, RSM 5E-mutant, $RPA32_{(172-270)}$, and $RPA70_{(1-120)}$, the lysate (denatured for RECQ4 proteins and native for RPA proteins) was applied to a HiTrap IMAC HP column charged with $CoCl_2$ (GE Healthcare) and step eluted in buffer without urea supplemented with 500 mM imidazole. The protein-containing fractions were incubated at 4 °C with 0.5 mg of TEV protease (produced in-house) per 1 L culture for 3 h at 25 °C to cleave the tag and then dialyzed overnight at 4 °C against 25 mM Tris-HCl [pH 8], 400 mM NaCl, 10 mM imidazole, and 2 mM BME. The dialyzed protein samples were passed through the same HiTrap IMAC HP column. The non-bound protein was collected, concentrated and applied on HiLoad 10/300 Superdex Increase 75 GL column (GE HealthCare).

$FLAG-APEX2-RECQ4_{(322-400)}$ was expressed in in Rosetta 2 (DE3) pLysS (71403; Novagen) *E. coli* strain for 3.5 h at 37 °C. The cells were lysed by sonication in a lysis buffer containing 50 mM Tris [pH 7.5], 1 M KCl, 0.1 M sucrose, 1 mM EDTA, 0.01% NP40, 1 mM BME and protease inhibitors. Clarified lysate was loaded on a column with Affi-Gel Blue Gel matrix (Biorad) and eluted with a gradient of 0–2.5 M NaSCN. Fractions containing $APEX2-RECQ4_{(322-400)}$ were pooled and dialyzed against 20 mM $K_2HPO_4$ [pH 7.5], 10% glycerol, 0.5 mM EDTA, 200 mM KCl, 0.01% NP40 and 1 mM DTT, followed by another dialysis against 20 mM $K_2HPO_4$ pH 7.5, 10% glycerol, 0.5 mM EDTA, 100 mM KCl, 0.01% NP40 and 1 mM DTT. The dialyzed sample was loaded on a Mono S column (GE HealthCare), equilibrated with 20 mM $K_2HPO_4$ [pH 7.5], 10% glycerol, 0.5 mM EDTA, 100 mM KCl, 0.01% NP40 and 1 mM DTT and eluted with a gradient of 0.1–1 M KCl. Fractions containing pure protein were concentrated on the VivaSpin2 column with 30 MWCO.

The human RPA heterotrimer expression vector (RPA70-RPA32-RPA14, UniProt codes: P27694, P15927, P35244) was a kind gift from Marc S. Wold. The heterotrimeric RPA complex and $MBP-RECQ4_{(1-400)}$-His were expressed and purified as previously described[56,140].

## Pull-down experiments

$MBP-RECQ4_{(1-400)}$-His and full-length RPA heterotrimer (8 µg of each) were incubated with Amylose Resin High Flow (E8022; NEB) for 30 min at 4 °C in 20 mM $K_2HPO_4$ [pH 7.5], 10% glycerol, 0.5 mM EDTA, 100 mM KCl, 0.01% NP40, and 2 mM BME. The unbound fraction was separated, the resin-bound fraction was washed, and fractions were analyzed by SDS-PAGE and Coomassie staining.

## DNA substrates preparation

Used DNA substrates are listed in Supplementary Table 1. DNA oligos were purchased from Sigma or Generi Biotech. For NMR and CD

measurements, they were purified using a HiLoad 10/300 Superdex Increase 75 GL column (GE HealthCare) equilibrated with milliQ water. DNA-containing fractions were collected, lyophilized and resuspended in an annealing buffer containing 25 mM KPO4 [pH 6.5] and 70 mM KCl. For oligo annealing, initial heating at 95 °C for 10 min was followed by slow overnight cooling at room temperature. For phase separation experiments oligos were dissolved in 20 mM KPO4 [pH 7.0] and 70 mM KCl, heated at 100 °C for 5 min, and then rapidly cooled down by putting on ice. Tetramolecular quadruplex (TetraG4) was folded from the single DNA oligonucleotide in 20 mM Tris-HCl [pH 7.5], 1 mM EDTA, and 1 M NaCl as previously described[123]. Concentrations were determined by spectrophotometry using extinction coefficients provided by the manufacturer.

## NMR spectroscopy

Experimental conditions (buffer composition and temperature) for NMR experiments shown in the manuscript are listed in Supplementary Table 6 according to the displayed item. For NMR and CD experiments shown in Fig. 5 that required high RSM stock concentration, a synthetic RSM peptide (sRSM: aa 358–388) was used (Peptide 2.0 Inc.). DNA binding and coacervation properties of RSM (aa 348–388) and sRSM (aa 358–388) did not differ (Supplementary Figs. 7b, c and 15a).

The RECQ4$_{(322–400)}$ sample used for chemical shift assignment contained ca. 0.7 mM, $^{13}$C- and $^{15}$N-labeled protein in 25 mM NaPO4 [pH 6.0], 100 mM NaCl, 2 mM BME, and 1 mM EDTA-d6 with 10% (v/v) $^2$H$_2$O added for the lock. Spectra were recorded at 298 K using a 700 MHz Bruker Avance II spectrometer equipped with a TXO cryogenic probehead with $z$ axis gradients. Protein backbone atom assignments were obtained using 3D CBCA(CO)NH, HNCACB, HNCA, HNCO, and HN(CA)CO experiments, and confirmed using a 4D HC(CC-TOCSY(CO))NH experiment[141,142]. The assignment completeness of RECQ4$_{(322–400)}$ exceeded 95% for all backbone chemical shifts (H$^N$, N, C$^α$, C$^β$, C′, H$^α$). The same set of 3D experiments was acquired along with a 3D HBHA(CBCACO)NH experiment to obtain the backbone RSM chemical shifts in complex with dsDNA or with RPA32C. NMR titrations of $^{15}$N-labeled proteins (RECQ4 or RPA) were typically performed using $^1$H-$^{15}$N HSQC experiments on samples containing 50 or 100 μM protein and step-wise addition of the binding partner. The assigned $^1$H-$^{15}$N HSQC spectrum of RSM is shown in Supplementary Fig. 8. Normalized chemical shift perturbations (CSPs) were calculated by using the equation: CSP = ([$δ_{HN}$]$^2$ + [$δ_N$/6]$^2$)$^{0.5}$. RPA32C assignments were taken from BMRB entry 4460. Spectra were analyzed using SPARKY[143].

G4 samples (50 μM) were diluted in 25 mM KPO4 [pH 6.5] and 70 mM KCl in the absence or in the presence of increasing RSM molar equivalents. G4 and RSM were pre-incubated at 25 °C for 5 min before NMR acquisitions. NMR spectra were acquired at 278, 288, or 298 K on Bruker Avance spectrometers equipped with cryogenic probes operating in frequencies ranging from 600 to 950 MHz. 1D experiments were registered, processed, and analyzed using Topspin4.0 (Bruker Biospin).

Standard relaxation experiments were utilized for measurements of longitudinal and transverse relaxation rates[144]. The longitudinal relaxation rate experiments were measured with: 0.0111, 0.0555*, 0.1221, 0.1887, 0.2664, 0.3552, 0.4662*, 0.5883, 0.7548, 0.9657, 1.2876*, 1.665 s relaxation delays and the transverse relaxation rate experiments were measured with 0.0, 14.4*, 28.8, 43.2, 57.6, 72.0, 86.4*, 100.8, 115.2*, 144.0*, 172.8 ms relaxation delays (asterisks denote spectra measured twice in all cases). The longitudinal and transverse relaxation rates were obtained using a fit of the decays of signal intensities to a mono-exponential decay function $I_{i,n,k} = A_{i,n}\exp(-R_{i,n} t_{i,k})$ in program Octave 3.8.2[145] using the function *leasqr* from the package *optim*. The index $i$ distinguishes variables related to the longitudinal ($i = 1$) and transverse relaxation ($i = 2$) rate measurement, index $n$ stands for data of $^{15}$N amide of an $n$-th residue, and index $k$ is related to

the measurement with the $k$-th relaxation delay. $I_{i,n,k}$ is the intensity of the signal read for a residue $n$ in a spectrum acquired with the $k$-th relaxation delay of the length $t_{i,k}$. $A_{i,n}$ and $R_{i,n}$ are optimized parameters, where the former is a pre-exponential factor and $R_{i,n}$ is the relaxation rate for the $n$-th residue. 2000 Monte-Carlo simulations of the measured signal intensities were then generated, and the fit was repeated for each of them to estimate the error of the relaxation rates. The obtained relaxation rates were recalculated to spectral density values using reduced spectral density mapping protocol[146–148] with the assumption that the high-frequency values are similar and their difference can be neglected.

Steady state NOE experiments were measured with a 6 s saturation period composed of inversion pulses separated by 11.1 ms[149]. The saturation irradiation was replaced by a 20 s long interscan delay for the measurement of the reference spectra. NOE parameters were calculated for each amide peak as a ratio of signal intensities read in the spectrum with saturation ($I_{sat}$) and in the reference ($I_{ref}$) spectrum NOE = $I_{sat}/I_{ref}$. The errors of the signal intensities were estimated based on the noise in the spectra in the regions with no peaks. The errors of NOE were calculated using the error propagation law.

## 2D lineshape analysis of competitive binding

$^1$H-$^{15}$N HSQC titration spectra were fitted using TITAN v1.6[108]. In the latest release, a competitive binding model (now available, including source code) was created to describe the interaction between free RSM protein, $P$, and two competing ligands $X$ and $Y$, representing RPA and DNA, to form the bound states $B_X$ and $B_Y$:

$$P + X \underset{k_{off}^X}{\overset{K_d^X}{\rightleftharpoons}} B_X \tag{1}$$

$$P + Y \underset{k_{off}^Y}{\overset{K_d^Y}{\rightleftharpoons}} B_Y \tag{2}$$

These equilibria were solved numerically for given total protein and ligand concentrations and dissociation constants to determine the concentrations of individual species, from which the exchange superoperator, representing exchange between free and ligand-bound spin states, was formed:

$$\frac{d}{dt}\begin{pmatrix} P \\ B_X \\ B_Y \end{pmatrix} = \begin{pmatrix} -k_{on}^X[X] - k_{on}^Y[Y] & k_{off}^X & k_{off}^Y \\ k_{on}^X[X] & -k_{off}^X & 0 \\ k_{on}^Y[Y] & 0 & -k_{off}^Y \end{pmatrix} \cdot \begin{pmatrix} P \\ B_X \\ B_Y \end{pmatrix} \tag{3}$$

where $k_{on}^X = k_{off}^X/K_d^X$ (4) and $k_{on}^Y = k_{off}^Y/K_d^Y$ (5). This was subsequently incorporated into the TITAN simulation routine, allowing titration spectra to be fitted to determine the model parameters $K_d^X$, $K_d^Y$, $k_{off}^X$ and $k_{off}^Y$. Four titration experiments were performed and combined to yield a total of 24 spectra that were then globally fitted, following standard protocols[109], using a selection of 16 representative residues. Reported uncertainties were determined using the jack-knife method.

## Electrophoretic mobility shift assay

Cy3-labeled tetramolecular G4 (40 nM), Cy3-labeled CEB1 G4 (30 nM) and/or FITC-labeled ssDNA 20mer (30 nM) were incubated with increasing concentration of RSM in 40 mM Tris-HCl [pH 7.5], 25 mM KCl, 5 mM MgCl$_2$, 2 mM DTT, 2% glycerol, and 0.1 mg/mL BSA for 10 min at 37 °C. To assess the dependence of RSM binding on ionic strength, FITC-labeled CEB1 G4 (30 nM) or FITC-labeled ds49 (15 nM) was incubated with RSM (320 nM or 1280 nM) in the same buffer supplemented with the indicated concentration of NaCl for 10 min at 37 °C. To compare the binding of different RECQ4 fragments, we incubated FITC-labeled ds49 (15 nM) with increasing concentrations of

RECQ4$_{(322-400)}$, RSM or sRSM for 10 min at 37 °C. To verify the folding of G4, BG4 protein (produced in-house) was incubated with tetramolecular G4 (40 nM) in 50 mM Tris-HCl [pH 7.5], 50 mM KCl, and 1 mM MgCl$_2$ for 10 min at 37 °C followed by stabilization of BG4-G4 complex by 0.01% glutaraldehyde for another 5 min at 37 °C. After the addition of loading buffer (60% glycerol, 10 mM Tris-HCl [pH 7.5], and 60 mM EDTA), resulting complexes were analyzed either on 10% polyacrylamide gel in 1× TBE (90 mM Tris-HCl [pH 8], 9 mM boric acid, and 0.2 mM EDTA) for 1 h at 95 volts (for tetramolecular G4, ds49 and ionic strength) or on 0.8% agarose gel in 1× TBE supplemented with 10 mM KCl for 40 min at 70 V at 4 °C (for CEB1 G4 and ssDNA 20mer). Gels were scanned on a FLA-9000 scanner (Fujifilm) or Typhoon™ laser-scanner (Cytiva) and quantified with Multi Gauge version 3.2 (Fujifilm). For the quantification of ionic strength experiment, we included two additional controls of the DNA substrate alone in 750 mM and 1000 mM NaCl because the fluorescence intensity of the DNA substrate could be affected by increasing ionic strength. These controls were used to normalize the DNA fluorescence intensity of the corresponding NaCl concentration in the presence of RSM. DNA fluorescence intensity at the lower NaCl concentrations was normalized to the fluorescence intensity without any additional salt.

## Fluorescence anisotropy measurements

Reactions containing 50 nM DNA substrate were incubated for 5–10 min at 25 °C with increasing concentrations of RSM and subsequently transferred to a 384-well microplate and read in a Tecan Microplate Reader Infinite F500 (Tecan group Ltd). Samples were excited with vertically polarized light at 490 nm, and both vertical and horizontal emissions were recorded at 525 nm. All measurements were conducted at 25 °C in 25 mM KPO4 [pH 6.5] and 70 mM KCl (low ionic strength) or 500 mM NaCl (high ionic strength). The data for each protein concentration was averaged over 10 min intervals to remove instrumental noise and processed by subtracting the anisotropy value obtained from the respective DNA substrate without the protein. Equilibrium dissociation constants ($K_d$) were calculated by fitting the data to the following equation:

$$FA = (([D] + [P] + K_d) - (([D] + [P] + K_d)^2 - (4 \times [D] \times [P]))^{1/2}) \times (A)/(2 \times [D]),$$

where [D] and [P] are concentrations of DNA and protein, respectively, and A is the maximum anisotropy value. Each data point is an average of three measurements.

## Circular dichroism spectroscopy

All spectra were collected from 185 to 330 nm with a spectral bandwidth of 1 nm for each sample using a Chirascan™ CD Spectrometer at room temperature (-20–25 °C). An average of four scans was recorded. To examine changes in the G4 signal due to RSM binding, the G4 concentration was fixed at 10 µM, and sRSM (aa 358–388) was added to yield the mentioned molar ratios. The samples of parallel/hybrid G4 were prepared in 25 mM KPO4 [pH 6.5] and 70 mM KCl, and the spectra were measured after overnight incubation of samples at 4 °C (experiments shown in Fig. 5 and Supplementary Fig. 14) or 5 min of incubation at room temperature (experiments shown in Fig. 6 and Supplementary Fig. 22). To evaluate CD spectra, values of maxima (265 nm for parallel G4 and 198 nm for sRSM) were determined. Centrifuged samples were spun for 2 min at 13,400 rpm and the supernatant was collected for the measurements. The CD spectrum of tetramolecular G4 (8 µM) was recorded in 40 mM Tris-HCl [pH 7.5], 25 mM KCl, 5 mM MgCl$_2$, 2 mM DTT, 2% glycerol, and 0.1 mg/mL BSA. The CD spectrum of 10 µM single DNA oligonucleotide (non-folded control) used to assemble the tetramolecular G4 was recorded in water.

## Isothermal titration calorimetry

Calorimetric titrations were carried out on an iTC200 microcalorimeter (MicroCal) at 30 °C. RSM and ligand (DNAs or RPA) samples were dissolved in 25 mM KPO4 [pH 6.5] and 70 mM KCl. The 300 µL sample cell was filled with a 0.1 mM solution of RSM, and the 40 µL injection syringe with 1 mM of the titrating ligand. Each titration consisted of a preliminary 0.2 µL injection followed by 19 subsequent 2 µL injections. The heat of the injections was corrected for the heat of dilution of every ligand into the buffer. Data fitting was done in Origin using the built-in one-site binding model.

## Microscopy of RECQ4 droplets

RECQ4 fragments at 10 µM final concentration were mixed with an equimolar amount of indicated DNA substrate (or 126 ng/µL for plasmid DNA - pBlueScript) in 25 mM KPO4 [pH 6.5] and 70 mM KCl, and transferred to a chambered slide (Ibidi) for imaging. Images were acquired at room temperature using a DeltaVision Elite microscope (GE Healthcare) equipped with a 60× oil-immersion objective. Fluorescence images were deconvolved with SoftWoRx software (GE Healthcare) and all images were finally processed in ImageJ. The fluorescence intensity of Thioflavin T inside the droplets was quantified manually by creating a line profile in ImageJ through the centre of each droplet and recording the maximum intensity value.

## Turbidity measurements

Increasing concentrations of RSM were mixed with 10 µM of the indicated DNA substrates in 25 mM KPO4 [pH 6.5] and 70 mM KCl, in a 96-well plate, and the optical density at 600 nm was measured on a Microplate Reader SpectraMax iD5 instrument immediately or after 5 min incubation at room temperature. For the ionic strength sensitivity experiment, increasing concentrations of NaCl (from 50 to 500 mM) were added to preformed droplets of sRSM-T95-2T and the optical density at 400 nm was measured.

## Stopped-flow assay

Stopped-flow experiments were measured using an SFM-3000 stopped-flow machine (Bio-Logic) equipped with a MOS-500 monochromator spectrometer (Bio-Logic) and an additional Photomultiplier control unit (PMS-250) for light scattering analysis. The intrinsic fluorescence of tryptophan of sRSM was recorded with an excitation wavelength of 295 nm and 320 nm cut-off filter. Light scattering was measured as a signal reading from a photomultiplier placed at an angle of 90° to the light source (at 295 nm). All experiments were performed at 25 °C in 25 mM KPO4 [pH 6.5] and 70 mM KCl. In the experiments with a fixed concentration of G4, 2.5 µM of T95-2T G4 was mixed with various concentrations of sRSM, ranging from 0 to 20 µM. Conversely, in the experiments with a fixed concentration of sRSM, 2.5 µM of sRSM was systematically mixed with increasing concentrations of T95-2T G4, ranging from 0 to 20 µM. When sRSM was fixed, fluorescence intensity together with light scatter were collected for 60 s or 180 s in the first and second replicated measurement, respectively, according to the following protocol: (a) every 0.001 s from 0-1 s; (b) every 0.01 s from 1–10 s; (c) every 0.1 s from 10–180 s. When G4 was fixed, fluorescence intensity together with light scatter were collected for 60 s or 150 s in the first and second replicated measurement, respectively, according to the following protocol: (a) every 0.001 s from 0–0.3 s; (b) every 0.01 s from 0.3–10 s; (c) every 0.1 s from 10–180 s. The curves shown are averages of 3 to 4 technical replicates.

## Spin-down assay

Indicated concentrations of sRSM and T95-2T G4 were mixed in 25 mM KPO4 [pH 6.5], 70 mM KCl and incubated at room temperature for 5 min. The samples were subsequently centrifuged for 3 min at 12,000 × $g$ (room temperature), supernatant was collected and the pellet resuspended in an equal volume of buffer. Both fractions were deproteinized with 2.5 µL proteinase K (10 mg/ml) and 2.5 µL 1% SDS, run on an 0.8% agarose gel in 1× TBE supplemented with 10 mM KCl for 40 min at 70 volts at 4 °C and stained by SYBR® Gold Nucleic Acid Gel

Stain (S11494, Thermo Fisher Scientific, diluted 1: 10,000 in 1x TBE). The gels were scanned using Typhoon™ laser-scanner, quantified using Multi Gauge version 3.2 (Fujifilm).

## Thioflavin T fluorescence quantification in solution

Thioflavin T (500 nM) was mixed with 10 μM of corresponding DNA substrates and where also indicated with 10 μM RSM in 25 mM KPO4 [pH 6.5] and 70 mM KCl. The fluorescence intensity at 490 nm (excitation wavelength: 445 nm) was measured by Microplate Reader SpectraMax iD5. The recorded values were normalized relative to the fluorescence of the Thioflavin T probe alone.

## G4 unwinding assay

The complex of RSM, MBP-RECQ4$_{(1-400)}$-His or APEX2-RECQ4$_{(322-400)}$ with tetramolecular G4 was assembled by incubation at 37 °C for 10 min in 40 nM Tris·HCl [pH 7.5], 25 mM KCl, 5 mM MgCl$_2$, 2 mM DTT, 2% glycerol, and 0.1 mg/mL BSA supplemented with 1 mM ATP and ATP regeneration system (Creatine Phosphate-Creatine Kinase). The G4 unwinding was initiated by adding ScPif1 helicase (3.1 nM or 6.3 nM) or FANCJ helicase (2.5 nM). After incubation at 37 °C for 15 min, the reaction was stopped by adding a mixture of 1% SDS and 10 mg mL$^{-1}$ proteinase K (A4392,0010, PanReac AppliChem) (1:1), resolved in 10% 1× TBE polyacrylamide gel at 95 volts for 1 h, and analyzed as above.

## Kinetic data analysis and statistics

The kinetic data were fit globally with the KinTek Explorer program (KinTek, USA), a dynamic kinetic simulation program that allowed multiple data sets to be fit simultaneously to a single model. Data fitting used numerical integration of rate equations from an input model (Fig. 6f) searching a set of parameters using the Bulirsch–Stoer algorithm with an adaptive step size that produces a minimum $\chi^2$ value calculated by using nonlinear regression based on the Levenberg-Marquardt method[150]. Residuals were normalized by sigma value for each data point.

The observable signal for circular dichroism data (CD) analyzed at 265 and 198 nm was defined as the sum of the contributions of G4 and RSM in soluble form (Equations 1 and 2), where $a_{265}$ and $a_{198}$ scales the signal to the concentration of G4 and sRSM-G4 at 265 and 198 nm, respectively and the factor $b_{198}$ scales the signal to the concentration of sRSM and sRSM-G4 at 198 nm. The observable signal for optical density measurements (OD) was defined as an approximate contribution of condensation products (Eq. 3), where $a_{OD}$ scales the signal to concentration and sensitivity of the measurement, the scaling factor $b_{n+1}$ defines the relative change in optical density contribution of the products of continual growth (sRSM-G4)$_{n+1}$ ($n \geq 3$). Similarly, the observable signal for light scattering measurements (LS) was defined as an approximate contribution of condensation products ($n \geq 3$) (Eq. 4), where $a_{LS}$ scales the concentration and $b_{n+1}$ defines the relative change in optical density contribution of the consecutive condensation species (RSM-G4)$_{n+1}$ ($n \geq 2$). The observable fluorescence signal was defined as the sum of the contributions of each species to the total fluorescence (Eq. 5), where $a_{FI}$ scales the signal to concentration and sensitivity of the measurement, factors $b_{FI}$ and $c_{FI}$ define the relative change in fluorescence in forming sRSM-G4 and (sRSM-G4)$_2$ complexes, respectively. Different scaling factors relating to the different sensitivity of analytical setups (factor $a$) were used for individual experiments, but the relative change in observable signal in forming individual complexes was constant for all datasets. The scaling factors were used as fitted parameters, and the obtained values are summarized in Supplementary Table 7.

$$Signal\,(CD_{265}) = a_{265} \cdot (G4 + sRSM - G4) \tag{6}$$

$$Signal\,(CD_{196}) = a_{198} \cdot (G4 + sRSM - G4) - b_{198} \cdot (sRSM + sRSM - G4) \tag{7}$$

$$Signal\,(OD) = a_{OD} \cdot ((sRSM - G4)_n + b_{n+1} \cdot (sRSM - G4)_{n+1} + \ldots + b_i \cdot (sRSM - G4)_i) \tag{8}$$

$$Signal\,(LS) = a_{LS} \cdot ((sRSM - G4)_n + b_{n+1} \cdot (sRSM - G4)_{n+1} + \ldots + b_i \cdot (sRSM - G4)_i) \tag{9}$$

$$Signal\,(FI) = 1 + a_{FI} \cdot sRSM - G4 + b_{FI} \cdot (sRSM - G4)_2 \tag{10}$$

The concentration dependence of NMR spectra was resolved by singular value decomposition (SVD) analysis integrated within KinTek Explorer. Extracted SVD amplitude vectors were then fitted to a proposed kinetic model by nonlinear regression analysis based on the numerical integration of rate equations. Alternating between the fitting of SVD amplitude vectors and the direct fitting of spectra helped us find the best global fit to the data. To account for slight variations in the data, sRSM or G4 concentrations were allowed to vary within an interval of ±10% to make the best fits possible. The standard error (s.e.) was calculated from the covariance matrix during nonlinear regression. The standard error estimates in fitted parameters were propagated to yield error estimates in calculated values, the equilibrium dissociation constant ($K_d$).

The free energy of binding was calculated for a reference state rather than the 1 M standard state (Eq. 11), where the product $K.[L] = [L]/K_d$ is a dimensionless number that defines the thermodynamic driving force for binding at the physiological concentration of ligand concentration[151]. The free energy profile was calculated using the Eyring equation (Eq. 12) at reference temperature 298 K, and reference concentration of 1 μM sRSM and 1 μM sRSM-G4.

$$\Delta G_0 = -R.T.\ln(K.[L]) \tag{11}$$

$$\Delta G^{\ddagger} = -R.T.\ln(k.h/k_B.T) \tag{12}$$

## Reporting summary

Further information on research design is available in the Nature Portfolio Reporting Summary linked to this article.

## Data availability

All the PDB codes cited in this paper (4MQV, 4GOP, 5N85, 2LK7, 2GKU) are available in the protein data bank web server. Chemical shift assignments of RPA32C were obtained from BMRB code 4460. The chemical shift assignments of RECQ4$_{(322-400)}$ can be accessed using the following accession code: BMRB 51487. All data generated in this study are available within the Article and Supplementary Information. Source data are provided with this paper.

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

## Acknowledgements

We thank Lukas Zidek for critical reading of the manuscript, Radovan Fiala for assistance with the NMR spectrometers and for photographing turbidity in the CD cuvette, Lukas Trantirek for helpful discussions, Klara Odehnalova for protein preparations, and Jaromír Toušek for excellent technical support. We also thank Pavel Janscak and Jean-Baptiste Boule for providing FANCJ and ScPif1 proteins, respectively. This research was supported by the Grant Agency of Masaryk University (MUNI/G/1594/2019) to K.T. and L.K. The K.T. laboratory was supported by the Czech Science Foundation (GA23-06913S). The L.K. laboratory was supported by the Czech Science Foundation (GACR 21-22593X) and Wellcome Trust Collaborative Grant (206292/E/17/Z). Z.P. was supported from the European Union's Horizon 2020 research and Innovation program TEAMING 857560, the Czech Ministry of Education, Youth and Sports TEAMING CZ.02.1.01/0.0/0.0/17_043/0009632, and Technology Agency of the Czech Republic (RETEMED TN02000122). P.K. was supported by European Regional Development Fund-Project MSCAfellow2@MUNI (No. CZ.02.2.69/0.0/0.0/18_070/0009846). C.A.W. was supported by the BBSRC (BB/T002603/1). K.K. was supported by CZ.02.01.01/00/22_008/0004575. CIISB, Instruct-CZ Centre of Instruct-ERIC EU consortium, funded by MEYS CR infrastructure project LM2023042, is gratefully acknowledged for the financial support of the measurements at the Josef Dadok National NMR Centre and at the CF Biomolecular Interactions and Crystallization.

## Author contributions

A.C.P., L.K. and K.T. conceived the project and designed the experiments. A.C.P. produced most of the protein samples, performed and analyzed NMR, FA, CD, and ITC experiments. M.P. performed the CD experiments, G4 unwinding assays, Stopped-Flow measurements and EMSAs. J.C. performed the pull-down experiment. Both M.P. and J.C. performed experiments describing coacervation. R.A. generated cell lines and performed the co-immunoprecipitation experiments. C.A.W. performed the 2D lineshape NMR analysis. P.K. measured and analyzed the NMR relaxation experiments. V.M. cloned and purified the APEX2-tagged RECQ4 fragment. K.K. assisted with NMR experiments. Z.P. constructed the global kinetic model and performed numerical simulations. L.K. and K.T. supervised the research. A.C.P. and K.T. wrote the paper. All authors discussed the results and commented on the manuscript.

## Competing interests

The authors declare no competing interests.
