## [Peer review file · Nature Communications]

REVIEWER COMMENTS

Reviewer #1 (Remarks to the Author):

Papageorgiou et al., demonstrated a novel function of positively charged IDR region, named RSM, of RECQ4. RSM was found to bind RPA in vivo and in vitro for the first time. NMR titration showed clearly the residues that are involved in formation of the RSM-RPA32C complex, which are basic residues in RSM and acidic residues in RPA32C. Then the same basic residues of RSM were shown by means of EMSA, FA, and NMR to be involved in binding with a various ssDNAs and dsDNAs. Distinct and competitive binding modes of RSM to DNA and RPA were demonstrated quantitatively by carrying out 2D NMR lineshape analysis. Disordered binding to DNA structure and disorder-to-order binding to RPA of RSM were shown with thermodynamics and kinetics data. ITC, CD, and NMR were effectively combined to obtain thermodynamical parameters to strengthen the predicted model, that RSM binds to folded G4s, remodels and traps them in situ in a dynamic heterogeneous ensemble of bound arrangements. By carrying out helicase assay, the authors confirmed that RSM binds and traps but does not unwind tetramolecular G4. In general, the experiments are solid, theoretically interesting, and conclusions reflect the data. However, a few issues need to be addressed or clarified before acceptance for publication.

L64: Is Supplementary Fig. 1a relevant here? It looks like Fig. 1a is relevant.

L81: Is “remodeled G4” in this occasion, G4 that is in an unfolded state?

L95: Could authors explain the advantages for using U2OS cells in the current study?

L101: Could authors describe the subunit and domain constitution of RPA? This referee found it difficult to figure out just by looking at Fig.1c.

L117: What is the completeness of the assignment?

L122: Is it “Fig. 1c and Fig. 2e”?

L124: Did the signals of tryptophan side chain HN(ϵ) of RSM perturb upon binding?

L128: Is the sequence of RSM similar to those of UNG2, XPA, SMARCAL1, and TIPIN?

L136: Can you comment on why the RPA32-RECQ4 5E interaction was observed at G1/S release 0 h? Additionally, why was the RPA32-RECQ4 5E interaction observed in unsynchronized conditions?

L144, L147, L152, L156: Are the sequences of bubble DNA, sprayed-arm DNA, dsDNA, and single-stranded DNA shown in the manuscript?

L162: Is the Supplementary table supplied in the file?

L167: There are two “bound” bands in the lanes 70, 140, and 210. Could the authors give a comment?

L177: Was the CSP of dsDNA imino proton also small? Did dsDNA imino proton CSP indicate unfolding of dsDNA?

L187: It is worthwhile to produce a “helical wheel” of the basic region of RSM and compare with those of other peptides. Are the basic residues buried in the cleft or stretched towards the acidic surface of the RPA32C? How about the two tryptophans in the basic region of RSM? Are they buried? Are the tryptophans involved in binding?

L214: 2D lineshape analysis seems very powerful. Could authors explain how k_{on} and k_{off} , and other parameters, if any, were obtained in the current case? Equations and/or a brief theory, and the parameters might be needed, may be as a supplementary information.

L232: Could authors explain how Fig. 4c, bottom was produced? Was it obtained from the ITC data?

L233: To rule out the possibility that the hydrophobic interaction is not involved especially in RPA-binding, the authors can show the “helical wheel” of the basic region.

L239: Indeed, disorder-to-order binding to RPA should be a low entropy event, however, shape-complementarity upon folding and binding may be advantageous in terms of water translational-entropy gain. Is it possible for RSM-DNA association to achieve water translational-entropy gain?

L248: Are the sequences of T95-T2 and HT shown in Fig. 5a and 5b?

L254: In Supplementary Fig. 9a, could the chemical shift perturbation be observed while signal intensity decreased? For example, did the tryptophan side chain HN(ϵ) of RSM perturb upon binding? If there is an indication of the involvement of tryptophan in binding, is it possible that the tryptophan has a role in unfolding of G4?

L269: The authors reasoned that the progressive reduction of the intensity of the imino protons is due to the loss of hydrogen bonding in the G4 tetrads upon unfolding of G-quadruplexes. However, is it possible that this is due to increased “heterogeneity caused by other process” as stated in L259?

L286: Did the authors monitor imino proton signals of dsDNAs (used in the previous section) in RSM titrating experiments? Does dsDNA unfold like G-quadruplex?

L294: Can authors clarify the procedure how to obtain Fig. 5c using equations and the exact data that has been used. Possibly in supplementary information? It seems that brief explanation is given in the Methods section, “Analysis of G4 binding thermodynamics”. However, the readers probably cannot follow the procedure. By the way, it seems that the reference in “L558” is Fig. 5c and not Fig. 4c.

L295: Can authors explain how the two fitting curves at the center of Fig. 5c. produced?

L296: Does G4 remodeling mean unfolding of G4 structure in this context?

L304: Does “remodels and traps” mean “unfolds and traps” in this context?

L306: Is the sequence of tetramolecular G4 given anywhere?

L308: The authors should explain why G4 unwinding cannot be exhibited by RSM is expected. This reviewer found that the usage of the terms, unwinding, unfolding, and remodeling, is not clear.

L309: In Supplementary Fig. 11d, "% DNA" values of both ssDNA and TetraG4 are constant around ca. 100, while in Supplementary Fig. 11e, "% DNA" values of ssDNA and TetraG4 add up to 100 in each SciPif1 concentration. Could authors describe the figure?

L310: What do authors meant by "remodel" the tetramolecular G4? Is it unwinding, unfolding, or another event? Is the RSM-bound and remodeled tetramolecular G4 in tetrameric state?

L312: In Fig. 6d, does the "G4 DNA" band contain just G4 DNA or G4 DNA and G4 DNA-RECQ4 complex?

L313: What exactly is the remodeled G4 in this context? Can the imino proton signals of the remodeled G4 be observed by 1D NMR? Is the remodeled G4 a mixture of parallel and anti-parallel G4s?

L318: What kind of G4 structural characteristics of tetramolecular G4 was lost by G4 remodeling? And what was maintained, if any?

L324: Could authors label G4F, RSM-G4F, and RSM-G4U peaks in Supplementary Fig. 9c?

L369: How does the functions of RSM correlate with those of the helicase and/or R4ZBD domains in RECQ4? Is there any speculation on how these domains function together in biological context?

L390: Some more information of shRNA sequence should be provided.

L499: "at at" → delete one "at"

L500: "Volts" → "volts"

L501: "gels visualized" → "gels were visualized"

L504: "2% Glycerol, 0.1 mg/ml BSA" → "2% glycerol, and 0.1 mg/mL BSA" (there are three corrections here)

L504 and below: The final two items in the list are usually separated by "and" or "or", which should be preceded by a comma. There are many in the Methods section, so please correct them.

L524: "*" → should be multiplication

L529: delete the "," preceding "."

L558: "Fig. 4c" → "Fig. 5c"

Reviewer #2 (Remarks to the Author):

This is an interesting and important study that adds significantly to the field and will have a noticeable impact. The manuscript is well-written and concise. There are several issues that should be addressed.

1) Abbreviations used in the abstract should be explained within the abstract text.

2) The authors stated that the Sld2-like region of RECQ4 is largely unstructured. This statement should be supported by the computational analysis. Even superficial analysis of the amino acid sequence of human RECQ4 (UniProt ID: O94761) using publicly available disorder predictors indicates that the first 475 residues of this protein are expected to be highly disordered.

3) As per bioinformatics analysis, RECQ4 is expected to undergo liquid-liquid phase separation. Therefore, it is likely that some phase separated liquid droplets can be formed by this protein alone or in the presence of DNA. It is highly recommended to check for the presence or absence of such droplets in the samples analyzed in this study.

Reviewer #3 (Remarks to the Author):

The authors study a positively charged disordered region within the human RECQ4 helicase protein, and show that it plays a role in remodeling and trapping G4-quadruplexes. The mechanism of action is studied primarily using NMR spectroscopy, complemented by other biophysical and cell-based studies. A two-step mechanism of action is described where the protein forms an encounter complex with the G-quadruplex and then remodels it using the inherent flexibility of the IDR. A second, mutually exclusive interaction with RPA is also described, which could indicate a regulatory role for the IDR requiring molecular hand-off.

The paper is a thorough exploration of the dynamics of interaction between the RECQ4 IDR domain and various forms of G4-quadruplexes. However, we have one major and some minor concerns/questions about the paper, detailed below. The materials and methods section requires more elaboration.

Major concern:

Figure 2f and Lines 130-137, Page 4: The authors state that the RECQ4 5E charge reversal mutant does not bind to RPA32C. While Supplementary Fig 3 does provide evidence of this through NMR, the IP panel

of Fig.2f shows binding with the 5E mutant at the initial time-point (synchronized panel 0). The authors should address this contradictory data.

Minor concerns/suggestions:

1) Page 2, line 56-57: The authors state that 'RECQ4 has a unique domain organization. It lacks the RQC and HRDC domains required for G4 unwinding by other members of the family..'. Fig. 1a shows that RQC is present in every RECQ protein but HRDC is not, so this sentence needs to be clarified – are both domains essential?

2) Page 2, line 64-65: 'The Sld2-like region is largely unstructured (Fig1 and supplementary Fig1a)'. It is not immediately clear how the conclusion of the region being unstructured is being drawn from these figures. Are the authors referring to the NMR chemical shifts? More elaboration is required in the text.

3) Page 4, line 117-129:

a. Line 119 – please edit to 'this allowed us to further map'

b. In Fig 2c please provide information on the multiple sequence alignment used to generate this data, and add the reference of the software used for analysis

4) Page 5, line 162: The Supplementary table mentioned in the text is missing

5) Page 7, line 223: What is the value of the Smolochowski constant being considered here, the size of the model being used, and does the model account for crowding?

6) Page 8, line 248: A figure depicting the structures of T59-2T and HT could be a helpful addition here

7) Page 9, line 291: Was a control experiment measuring the CD spectrum of just the RSM peptide done, and could that be added to the panel showing the comparative CD spectra to ensure that the changes in the signal are only due to the interactions between RSM and the DNA and not due to the presence of RSM itself.

8) Materials and Methods section:

a. Please add accession codes for proteins used

b. Page 12, line 387 and line 420: Please provide plasmid maps/ more information on cloning sites

c. Page 12, line 412: Please provide details of antibodies used – company, catalog number etc. Also details of the chemiluminescence using HRP substrate – if this was a kit, please mention which one.

d. Page 13, line 439: Please provide primer sequences for cloning and expression protocol for RSM.

e. Page 13, line 457: Were the DNA substrates checked for depurination after heating for 10 min at 95C?

f. Page 15, line 498: What was the label used for DNA?

9) Supplementary Figures:

- a. Supp. Fig.10: Please include the polydispersity index of the peaks
- b. Supp. Fig.11, panel e: The SD is not present or not visible.
- c. Supp. Fig.12: Are the quantification panels an average of two runs? Please clarify.

Point-by-point response

We would like to thank the reviewers for their insightful comments, especially regarding possible phase separation properties. In response to their valuable feedback, we set out a comprehensive set of approaches to elucidate whether RSM complexation with G-quadruplexes is accompanied by phase separation and droplet formation. The following points outline our efforts to address these concerns:

- 1) Optical microscopy validation: We have demonstrated the rapid droplet formation upon the mixing of RSM with G4s, a phenomenon that is absent when these components are analyzed alone (Fig. 5c,d and Supplementary Fig. 14c).
- 2) Confirmation by fluorescence microscopy: To further validate the presence of both RSM and G4 within these droplets, we have employed fluorescently labelled components, effectively confirming their coexistence within the droplets (Fig. 5d,e).
- 3) Specificity and variability: We extended our investigation to encompass diverse binding partners, confirming that droplet formation is observed with parallel and hybrid G4s, and varying lengths of ssDNA, but not with other interacting molecules including dsDNA or RPA32C (Supplementary Fig. 14c).
- 4) Complex coacervation: The process of droplet formation is demonstrated to follow a complex coacervation mechanism, which is dependent on charged polyelectrolytes [10.1039/d0sm00001a], as RSM-DNA droplets dissolve in a salt-dependent manner (Fig. 5c). This is further supported by the behavior of the 5E-mutant that effectively neutralizes the charge within RSM, resulting in the loss of observable droplets while residual binding for parallel G4 is still detectable by NMR (Supplementary Fig. 14d,e).
- 5) Reinterpretation of CD and NMR data: Based on these new insights, we revisited our initial interpretation of the loss of G4-specific CD and NMR signals, now attributing it to droplet formation rather than G4 remodeling. We substantiated this by new data revealing that RSM-DNA droplets are optically inactive (Supplementary Figure 16). This data represents a novel observation that has not been previously reported for other droplets. Our present understanding indicates that the decrease in G4 signal upon RSM titration reflects the degree of coacervation, and that CD cannot be used to assess the G4 structure inside these droplets. Similarly, the decrease in G4 imino peaks upon RSM binding is explained by droplet formation, while the chemical shift perturbations represent the initial RSM-G4 encounter complex.
- 6) G4 structure within droplets: We have also probed the status of G4 structure inside the droplets using Thioflavin T, a fluorescent dye that displays increase fluorescence upon binding various G4 structures compared to ssDNA [10.1093/nar/gku111]. Our new data demonstrate an increase of Thioflavin T fluorescence in a sample containing RSM-G4 droplets, suggesting that G4s remain folded upon coacervation with RSM (Fig. 7).
- 7) Further verification of phase separation behavior: To ensure the reproducibility and validity of our findings concerning the phase separation, we repeated NMR experiments with G-quadruplexes in the presence of two RECQ4 peptides (RSM(348-388) and sRSM(358-388); Supplementary Fig. 15).
- 8) Rectification of data misinterpretation: We have scrutinized our data and identified and advertent misinterpretation in the DLS data caused by unintentional centrifugation before the measurements, resulting in droplets removal. Consequently, we have removed these data from the manuscript.
- 9) Distinguishing coacervates from G4 trapping: Interestingly, our data also indicate that the coacervation phenomenon is not responsible for the observed inhibition of Pif1 helicase activity on tetramolecular G4, an event previously described as “G4 trapping” in the original manuscript. This is substantiated by the behavior of the Apex-RECQ4

(322-400) fragment, which still binds G4 and blocks Pif1 yet does not undergo droplet formation (Supplementary Fig. 22).

- 10) Global numerical model of coacervation: To provide a comprehensive understanding, we have integrated several experimental readouts (CD, NMR, stop-flow, turbidity measurements, and light scattering) into a global numerical model describing the coacervation in our system, including its kinetics. This model reveals a rapid nucleation-growth mechanism leading to a dynamic equilibrium between the dilute and condensed phases (Fig. 6). The predictions derived from our model were further validated by a spin-down assay, monitoring the G4 fraction in the dilute phase (Fig. 6i).

We believe that the revised version of our manuscript provides a corrected interpretation of our data. This revision not only enhances the quality of our study but also opens new and fascinating questions for RECQ4 biology and G4-modulated phase separation. Based on our data, we speculate that the engagement of the disordered RSM with G4 represents a barrier for processing enzymes. An attractive working hypothesis is that RSM could act as a phase-separating element that impacts the cellular functions of RECQ4 in a fashion similar to IDRs of DNA replication initiators that drive DNA-dependent phase separation [10.7554/eLife.70535]. This proposed function propels new dimension for understanding RECQ4's role, particularly in the telomere and mitochondrial DNA maintenance, and repair of DNA double-strand breaks, all of which are directly influenced by G4 structures.

In response to the reviewer's valuable feedback, we have meticulously addressed all the points raised. Our commitment to providing a comprehensive response is reflected in the additional experiments performed, the creation of two new Figures, eleven Supplementary Figures and six Supplementary Tables. Moreover, we have revisited and revised the "Binding and coacervation between RSM and G-quadruplexes" section in the Results, along with the Discussion, to accurately portray our new insights into phase separation. To facilitate easy orientation, the revised text is highlighted in grey throughout the revised manuscript, and our responses to the reviewers' comments are indicated in red font color. We always refer to the text line numbers and the items of the Supplementary materials (Figures and Tables) as they appear in the revised manuscript.

We believe these revisions not only address the reviewers' concerns but also elevate the significance and novelty of our study. We appreciate the opportunity to refine our manuscript and eagerly await the reviewers' assessment of our revised work.

Reviewer #1 (Remarks to the Author):

Papageorgiou et al., demonstrated a novel function of positively charged IDR region, named RSM, of RECQ4. RSM was found to bind RPA in vivo and in vitro for the first time. NMR titration showed clearly the residues that are involved in formation of the RSM-RPA32C complex, which are basic residues in RSM and acidic residues in RPA32C. Then the same basic residues of RSM were shown by means of EMSA, FA, and NMR to be involved in binding with a various ssDNAs and dsDNAs. Distinct and competitive binding modes of RSM to DNA and RPA were demonstrated quantitatively by carrying out 2D NMR lineshape analysis. Disordered binding to DNA structure and disorder-to-order binding to RPA of RSM were shown with thermodynamics and kinetics data. ITC, CD, and NMR were effectively combined to obtain thermodynamical parameters to strengthen the predicted model, that RSM binds to folded G4s, remodels and traps them in situ in a dynamic heterogeneous ensemble of bound arrangements. By carrying out helicase assay, the authors confirmed that RSM binds and traps but does not unwind tetramolecular G4. In general, the experiments are solid, theoretically interesting, and conclusions reflect the data. However, a few issues need to be addressed or clarified before acceptance for publication.

L64: Is Supplementary Fig. 1a relevant here? It looks like Fig. 1a is relevant.

Correct, only Fig. 1a was relevant in the original version of the manuscript. However, based on the comments of the other reviewers, we have prepared a new “Supplementary Fig. 1” where sequence and AlphaFold2 analysis demonstrates better the disorderedness of the Sld2-like region of RECQ4. In L73 of the revised manuscript, we now refer to the new “Supplementary Fig. 1”.

L81: Is “remodeled G4” in this occasion, G4 that is in an unfolded state?

Based on the phase separation findings listed above, the terms “remodel” and “unfold” are not used throughout the revised manuscript.

L95: Could authors explain the advantages for using U2OS cells in the current study?

U2OS is a standard mammalian cell line used for studying various aspects of biology. It provides the possibility to use an isogenic cell line (U2OS Flp-In T-Rex), allowing controlled downregulation of endogenous RECQ4 and inducible expression of exogenous WT or mutant version of RECQ4.

L101: Could authors describe the subunit and domain constitution of RPA? This referee found it difficult to figure out just by looking at Fig.1c.

A new “Supplementary Fig. 2” has been added, describing the subunit and domain organization of RPA in detail. The following concise description has also been added in the revised manuscript (L103-L106). “RPA is a protein composed of three subunits (RPA70, RPA32, RPA14) and exhibits a modular architecture^{92,93} that allows globular domains to associate in a trimeric core (70C, 32D, 14), bind ssDNA (70A, 70B, 70C, 32D), or participate in protein interactions (RPA70N, RPA32C) (Supplementary Fig. 2).”

L117: What is the completeness of the assignment?

“The assignment completeness of RECQ4₍₃₂₂₋₄₀₀₎ was 95% for all backbone chemical shifts (HN, N, CA, CB, CO, HA) and 80% for all-atom chemical shifts.” This information has been added in the Methods section of the revised manuscript (L605-L606). The chemical shifts of RECQ4₍₃₂₂₋₄₀₀₎ have been deposited to BMRB, as stated in the Data availability section (L833-L836). The backbone chemical shifts measured to derive the structural and dynamic descriptors of RSM (aa 348-388) free, in complex with dsDNA, or complex with RPA, are given in Supplementary Tables 2-4.

L122: Is it “Fig. 1c and Fig. 2e”?

We have corrected the text in the revised manuscript (L130).

L124: Did the signals of tryptophan side chain HN(ϵ) of RSM perturb upon binding?

The tryptophan side chains do not show large CSPs upon binding to RPA, suggesting no contribution to RPA binding. Fig. 2a and Fig. 3a have been updated and include the binding trajectories of the tryptophan sidechains in the serial titrations with RPA or dsDNA.

L128: Is the sequence of RSM similar to those of UNG2, XPA, SMARCAL1, and TIPIN?

A new “Supplementary Fig. 5” depicting a sequence alignment of the RPA32C interaction motifs in UNG2, XPA, SMARCAL1, TIPIN, and RSM has been added and cited in L137 of the revised manuscript. Positively charged and hydrophobic residues conserved at specific positions are highlighted.

L136: Can you comment on why the RPA32-RECQ4 5E interaction was observed at G1/S release 0 h? Additionally, why was the RPA32-RECQ4 5E interaction observed in unsynchronized conditions?

Detected residual interaction between RECQ4 5E-mutant and RPA could be explained by other domain(s) contributing to the interaction within the full-length context of these proteins. In response to a similar comment from another reviewer, we state in L147-L148 of the revised manuscript that “the basic patch of RSM motif is the principal determinant of RPA-RECQ4

interaction (Fig. 2f)". Furthermore, we also added a sentence in discussion (L434-L436) stating: "However, we cannot exclude the possibility that the RSM-RPA32C interaction may be stabilized by other regions in the context of the RPA heterotrimer."

We do not believe there is an interaction between RPA and RECQ4 detected in unsynchronized conditions since the RPA32 signal is comparable to the control EGFP lane, representing background RPA binding.

L144, L147, L152, L156: Are the sequences of bubble DNA, sprayed-arm DNA, dsDNA, and single-stranded DNA shown in the manuscript?

They are now included in Supplementary Table 1.

L162: Is the Supplementary table supplied in the file?

Thank you for this remark. The table was accidentally removed during the manuscript submission but now is present as "Supplementary Table 1" with all relevant information.

L167: There are two "bound" bands in the lanes 70, 140, and 210. Could the authors give a comment?

The bubble DNA substrate contains two ds/ss DNA junctions that are preferred RSM binding sites. Therefore the two bands observed in the EMSA experiment likely represent 1:1 and 2:1 RSM:DNA complexes.

However, for simplicity and clarity, we have decided to remove the data related to DNA bubble binding in the revised manuscript. Instead, we analyzed and compared RSM binding to ssDNA, dsDNA, parallel and hybrid G4 (Fig. 3c) since these substrates were tested for coacervation (Fig. 5 and Supplementary Fig. 14) and salt-dependent binding (Fig. 3d and Supplementary Fig. 7f).

L177: Was the CSP of dsDNA imino proton also small? Did dsDNA imino proton CSP indicate unfolding of dsDNA?

In dsDNA titration with RSM, the dsDNA imino signals experience chemical shift perturbations due to binding. Compared to RSM amide CSPs, the dsDNA imino CSPs are smaller in the proton dimension. The fact that the imino protons do not experience severe broadening and retain their signal intensity in the presence of excess RSM, strongly suggests that they remain hydrogen bonded in the double helix upon RSM binding. The new experimental data are depicted in the new "Supplementary Fig. 7d", which is cited in L170-171 of the revised manuscript: "the binding of RSM to dsDNA did not affect the integrity of the DNA helix (Supplementary Fig. 7d)"

L187: It is worthwhile to produce a "helical wheel" of the basic region of RSM and compare with those of other peptides. Are the basic residues buried in the cleft or stretched towards the acidic surface of the RPA32C? How about the two tryptophans in the basic region of RSM? Are they buried? Are the tryptophans involved in binding?

We have prepared a sequence alignment of peptide segments from different proteins known to interact with RPA32C in an induced helix conformation. The conserved cationic residues are highlighted. Based on the NMR analysis, we generated a helical wheel for the RSM residues with strong helical propensity and superimposed it on the SMARCAL1 helix of the RPA32C-SMARCAL1 complex (PDB: 4mqv). The model suggests that 6 out of the 8 positively charged residues of the RSM helical wheel project like antennas on the sides of the helix and are in direct contact with the acidic cleft of RPA32C. In contrast, the model predicts that the two tryptophan sidechains protrude from the opposite site of the RPA32C cleft, suggesting they are not involved in binding. To test this, we generated a double tryptophan mutant W379A/W383A and monitored RPA32C binding by NMR. The fingerprint spectra (WT vs W379A/W383A) show a very similar interaction, as many RSM residues experience the same trajectories upon RPA32C interaction. The analysis the reviewer requested and the new experimental data are shown in new "Supplementary Fig. 5" cited in L137-L140 of the revised manuscript, where we added: "Unlike the other RPA partners, RECQ4 contains two well-

conserved tryptophans in the binding motif. However, an RSM double tryptophan mutant (W379A/W383A) did not affect RPA32C binding in vitro as judged by the CSP binding profile (Supplementary Fig. 5b, c).”

L214: 2D lineshape analysis seems very powerful. Could authors explain how k_{on} and k_{off} , and other parameters, if any, were obtained in the current case? Equations and/or a brief theory, and the parameters might be needed, may be as a supplementary information.

We have added a brief explanation of the method in the main text (L231-L234 of the revised manuscript) and a complete description of the competitive binding model in the methods section (L641-L656 of the revised manuscript).

L232: Could authors explain how Fig. 4c, bottom was produced? Was it obtained from the ITC data?

Yes, the thermodynamic parameters of the binding were obtained from the ITC data displayed in the new “Supplementary Fig. 11”. In an ITC experiment, we are interested in the integrated heats of injection that directly report on the ΔH of a binding event and allow us to extract K_a from a curve fit of the integrated heat. We can then extract other parameters (ΔG and $-T\Delta S$) by the knowledge that $\Delta G = -RT\ln K_a$ and $\Delta G = \Delta H - T\Delta S$, where R is the gas constant and T is the absolute temperature of the experiment.

L233: To rule out the possibility that the hydrophobic interaction is not involved especially in RPA-binding, the authors can show the “helical wheel” of the basic region.

The sequence alignment and the helical wheel are shown in the new “Supplementary Fig. 5a, b”. While hydrophobic interactions are also present, e.g., the conserved hydrophobic dipeptide, they could not account for the large entropy difference measured because, in both interactions (RSM with RPA and dsDNA), the RSM cationic residues are the main contributors to binding.

L239: Indeed, disorder-to-order binding to RPA should be a low entropy event, however, shape-complementarity upon folding and binding may be advantageous in terms of water translational-entropy gain. Is it possible for RSM-DNA association to achieve water translational-entropy gain?

Without performing ITC measurements across multiple temperatures to assess changes in the heat capacity, which we feel is beyond the scope of this study, it is difficult to identify such effects. However, it is worth noting that we are comparing two different binding modes of the same sequence (RSM). While there are other differences in the two interactions, to some level of approximation, the change in solvent-accessible surface area is similar. Therefore, the effect on solvent translational entropy will (at least partially) cancel out when comparing the two interactions.

L248: Are the sequences of T95-T2 and HT shown in Fig. 5a and 5b?

Yes, they are shown as follows:

Parallel G4: 5'-TT(GGGT)₄-3' (T95-T2)

Hybrid G4: 5'-TT(GGGTTA)₃GGGA-3' (HT)

First, we provide information on the topology (Parallel or Hybrid), the corresponding sequences, and the acronyms in brackets as reported in the original studies. According to the request of another reviewer, we have also added a new “Supplementary Fig. 12” cited in L269 of the revised manuscript, with the structures and sequences of the two quadruplexes used in our studies.

L254: In Supplementary Fig. 9a, could the chemical shift perturbation be observed while signal intensity decreased? For example, did the tryptophan side chain HN(epsilon) of RSM perturb upon binding? If there is an indication of the involvement of tryptophan in binding, is it possible that the tryptophan has a role in unfolding of G4?

Indeed, CSPs are observed at a 1RSM:0.5G4 ratio. Still, they are too small because they report on the weighted average of the RSM-G4 complex in the dilute phase, which represents

~5% of the RSM population in solution as the majority of the RSM-G4 complex has partitioned to the condensed phase that is not observed in solution NMR. The tryptophan side chains follow the same trend as the main-chain peaks.

We tested the double tryptophan mutant (W379A/W383A) for the ability to form droplets and we did not observe any noticeable difference in coacervation with G4s. These data are shown in the new “Supplementary Fig. 14” and cited in L294-L296 of the revised manuscript: “On the other hand, the RSM double tryptophan mutant was prone to forming droplets with both G4s, suggesting that the two tryptophan residues within the RSM are not critical for coacervation (Supplementary Fig. 14d, e).”

L269: The authors reasoned that the progressive reduction of the intensity of the imino protons is due to the loss of hydrogen bonding in the G4 tetrads upon unfolding of G-quadruplexes. However, is it possible that this is due to increased “heterogeneity caused by other process” as stated in L259?

We thank the reviewer for this critical remark. Indeed, signal intensity reduction of the imino protons is due to the formation of droplets that contribute no signal due to enhanced relaxation rates in the condensed phase. We now state in L277-L278 of the revised manuscript: “However, the fact that RSM peaks synchronously lose intensity (Supplementary Fig. 13a, b) is indicative of the possible formation of high-order complexes.”

L286: Did the authors monitor imino proton signals of dsDNAs (used in the previous section) in RSM titrating experiments? Does dsDNA unfold like G-quadruplex?

We performed these experiments, and the data monitoring the CSPs of the imino protons of dsDNA, and thus RSM binding, are shown in the new “Supplementary Fig. 7d”.

L294: Can authors clarify the procedure how to obtain Fig. 5c using equations and the exact data that has been used. Possibly in supplementary information? It seems that brief explanation is given in the Methods section, “Analysis of G4 binding thermodynamics”. However, the readers probably cannot follow the procedure. By the way, it seems that the reference in “L558” is Fig. 5c and not Fig. 4c.

We acknowledge that the procedure for generating Fig. 5c and its underlying equations might not have been adequately elucidated in our previous version. In light of the new findings, we now describe new three-step model of charge-driven coacervation by integrating equilibrium and kinetic binding data in a global numerical model. Data analysis and the interpretation of the coacervation model are summarized in new Fig. 6, in new Supplementary Fig. 17 and 18, in a new Supplementary note, in new Supplementary Table 7, in L326-L363 of the revised manuscript. Furthermore, the revised Methods section now includes a more comprehensive account of the experiments and details (L779-L831) elucidating the derivation of the numerical model.

L295: Can authors explain how the two fitting curves at the center of Fig. 5c. produced?

Fitted parameters of data displayed in new Fig. 6 and used to derive the numerical model of coacervation are described in L779-L831 of Methods and listed in Supplementary Table 7.

L296: Does G4 remodeling mean unfolding of G4 structure in this context?

In light of the phase separation findings, the terms “remodel” and “unfold” are not used anymore in the revised manuscript.

L304: Does “remodels and traps” mean “unfolds and traps” in this context?

As explained earlier, the terms “remodel”, “unfold”, and “trap” are not used anymore in the revised manuscript.

L306: Is the sequence of tetramolecular G4 given anywhere?

The sequence of the single oligonucleotide that folds into tetramolecular G4 after annealing is given in “Supplementary Table 1”.

L308: The authors should explain why G4 unwinding cannot be exhibited by RSM is expected. This reviewer found that the usage of the terms, unwinding, unfolding, and remodeling, is not clear.

We apologize for the confusion. We have removed “as expected” in L388 of the revised manuscript to state that RSM does not unwind G4s.

Concerning the terminology, we now use only G4 unwinding/unfolding to refer to the dissociation of G4 structures resulting in the formation of single-stranded DNA. For example, unwinding the tetramolecular G4 by Pif1 results in single-stranded DNA, as shown in the new “Supplementary Fig. 20c, e”.

L309: In Supplementary Fig. 11d, “% DNA” values of both ssDNA and TetraG4 are constant around ca. 100, while in Supplementary Fig. 11e, “% DNA” values of ssDNA and TetraG4 add up to 100 in each SciPif1 concentration. Could authors describe the figure?

For clarity, both panels in the new Supplementary Fig. 20 (panels d and e) now depict the percentage of individual DNAs (tetramolecular G4 DNA and ssDNA) from input DNA containing no protein. We have modified the new “Supplementary Fig. 20” legend accordingly.

L310: What do authors meant by “remodel” the tetramolecular G4? Is it unwinding, unfolding, or another event? Is the RSM-bound and remodeled tetramolecular G4 in tetrameric state?

As explained earlier, the terms “remodel” and “unfold” are not used anymore in the revised manuscript.

RSM binds the tetramolecular G4, which becomes inaccessible to other helicases in this complex. The tetramolecular G4 likely stays in a tetrameric state while in complex with RSM since the removal of RSM from the complex by Proteinase K does not affect the G4 structure (Fig. 8a) (e.g., the ratio between tetramolecular G4 and ssDNA does not change as it happens with Pif1 that unwinds the tetramolecular G4 to its ssDNA strands (Supplementary Fig. 20c)).

We also addressed the G4 status inside the RSM-DNA droplets using Thioflavin T, whose fluorescence increases upon binding to various G4 structures compared to ssDNA. Our new data demonstrate an increase of Thioflavin T fluorescence of a sample containing RSM-G4 droplets, suggesting that G4s remain folded upon coacervation with RSM (Fig. 7d).

L312: In Fig. 6d, does the “G4 DNA” band contain just G4 DNA or G4 DNA and G4 DNA-RECQ4 complex?

The “G4 DNA” band corresponds only to G4 DNA since all proteins have been removed by proteinase K digestion before loading on the gel (Fig. 8 in the revised manuscript).

L313: What exactly is the remodeled G4 in this context? Can the imino proton signals of the remodeled G4 be observed by 1D NMR? Is the remodeled G4 a mixture of parallel and anti-parallel G4s?

We did not study the tetramolecular G4 by NMR to examine the behavior of the imino signals. The CD signature of the free tetramolecular G4 (Supplementary Fig. 20b) is indicative of a parallel orientation of its strands. Our data do not indicate that RSM binding could affect the G4 strand orientation and G4 structure.

As stated above, using Thioflavin T, whose fluorescence increases upon binding to various G4 structures compared to ssDNA, we also addressed the G4 status inside the RSM-DNA droplets. These new data demonstrate an increase of Thioflavin T fluorescence of a sample containing RSM-G4 droplets, suggesting that G4s remain folded upon coacervation with RSM (Fig. 7d).

L318: What kind of G4 structural characteristics of tetramolecular G4 was lost by G4 remodeling? And what was maintained, if any?

There is no loss of structural characteristics by G4 binding. We know now that the loss of the tetramolecular G4 characteristic CD signal is due to the partition in the condensed phase.

L324: Could authors label G4F, RSM-G4F, and RSM-G4U peaks in Supplementary Fig. 9c?

The suggested labeling is only feasible for the cyan spectrum in the new Supplementary Fig. 13c, representing the free G4. RSM-G4 complex in the dilute phase is evidenced in the orange spectrum by the CSP of the G4 imino peaks representing the weighted average of the populations between G4 free and RSM-G4 complex in solution. RSM-G4 complex in the condensed phase is not visible due to enhanced relaxation rates in the condensed phase leading to extreme signal broadening.

L369: How does the functions of RSM correlate with those of the helicase and/or R4ZBD domains in RECQ4? Is there any speculation on how these domains function together in biological context?

We thank the reviewer for the fascinating question. The helicase-R4ZBD module could be involved in processing dsDNA, as shown [10.1038/ncomms15907], whereas RSM could exert a signaling role related to G4 presence. However, we cannot rule out that the engagement of the helicase domain might provide the energy for destabilizing the RSM/G4 complex. A potential RNA binding motif in RECQ4 [10.1038/srep21501] might also represent a possible function in the binding and processing of R-loops. These structures are known to contain both RNA and G4 components [10.1093/nar/gkaa1206]. All these hypotheses are currently being tested. The great challenge would be to establish whether full-length RECQ4 and G4 structures are found together in phase-separated structures in the cell.

L390: Some more information of shRNA sequence should be provided.

The shRNA target sequence (5'-TAGGAAGAGCCTCATCTAAG-3') is given in L473 of the revised manuscript.

L499: "at at" → delete one "at"

We apologize for these errors, which have been corrected.

L500: "Volts" → "volts"

Thank you, it has been corrected (L671, L673, L759, L776 in the revised manuscript).

L501: "gels visualized" → "gels were visualized"

Thank you, we now state that "Gels were scanned on a FLA-9000 scanner (Fujifilm) or Typhoon™ laser-scanner (Cytiva)" (L673-L674 in the revised manuscript).

L504: "2% Glycerol, 0.1 mg/ml BSA" → "2% glycerol, and 0.1 mg/mL BSA" (there are three corrections here)

Thank you, it has been corrected (L708-L709 in the revised manuscript).

L504 and below: The final two items in the list are usually separated by "and" or "or", which should be preceded by a comma. There are many in the Methods section, so please correct them.

Thank you, we did our best to correct as many instances as possible.

L524: "*" → should be multiplication

Thank you, it has been corrected (L693 in the revised manuscript).

L529: delete the "," preceding "."

Thank you, it has been corrected.

L558: "Fig. 4c" → "Fig. 5c"

This figure is not anymore part of the revised manuscript and has been replaced by the numerical analysis shown in new Fig. 6.

Reviewer #2 (Remarks to the Author):

This is an interesting and important study that adds significantly to the field and will have a noticeable impact. The manuscript is well-written and concise. There are several issues that should be addressed.

1) Abbreviations used in the abstract should be explained within the abstract text.

We have provided the full protein names in the abstract: “ATP-dependent DNA helicase Q4 (RECQ4)” and “Replication protein A (RPA)”.

2) The authors stated that the Sld2-like region of RECQ4 is largely unstructured. This statement should be supported by the computational analysis. Even superficial analysis of the amino acid sequence of human RECQ4 (UniProt ID: O94761) using publicly available disorder predictors indicates that the first 475 residues of this protein are expected to be highly disordered.

We have included the sequence analysis and AlphaFold2 structure prediction in the new “Supplementary Fig. 1”, suggesting that the Sld2-like region of RECQ4 is mainly disordered. The new “Supplementary Fig. 1” is included in the revised manuscript and cited in L73.

3) As per bioinformatics analysis, RECQ4 is expected to undergo liquid-liquid phase separation. Therefore, it is likely that some phase separated liquid droplets can be formed by this protein alone or in the presence of DNA. It is highly recommended to check for the presence or absence of such droplets in the samples analyzed in this study.

We are profoundly grateful for your astute recommendation, which has undoubtedly elevated the quality and rigor of our study. Your insightful recommendation prompted us to delve into this aspect, and our investigations have led to some significant insights.

Upon conducting optical microscopy analyses, we have indeed confirmed the presence of phase-separated droplets resulting from the complexation of RSM with G4 structures and ssDNA. In contrast, RSM alone or in complex with dsDNA did not show signs of phase separation. The discernible sensitivity of RMS-G4 droplets to salt concentration implies that they are formed by charge-driven complex coacervation between the positively charged RSM and the negatively charged G4. In response to this revelation, we performed a series of new experiments (including microscopy, turbidity measurements, CD, NMR and stopped-flow analyses) to understand this phenomenon and integrate it with existing data. Notably, we found that CD cannot be used to assess the state of G4 inside the RSM droplets since these are optically inactive. This led us to reinterpret the loss of G4-specific CD and NMR signal upon addition of RSM. Instead of the originally proposed G4 remodeling we now show that the signal loss reflects the transition of G4 into the condensates. Finally, based on these new findings, we have developed a comprehensive global numerical model describing the kinetics of RSM-G4 coacervation. This model serves to include our current understanding and provide a mechanistic framework to comprehend the intricate interplay of the components involved.

Thank you very much again for pointing out the possibility of phase separation in our system. Your comment directed us to perform additional key experiments, correctly interpret our existing data but also paved the way to significantly improve the manuscript.

Reviewer #3 (Remarks to the Author):

The authors study a positively charged disordered region within the human RECQ4 helicase protein, and show that it plays a role in remodeling and trapping G4-quadruplexes. The mechanism of action is studied primarily using NMR spectroscopy, complemented by other biophysical and cell-based studies. A two-step mechanism of action is described where the protein forms an encounter complex with the G-quadruplex and then remodels it using the inherent flexibility of the IDR. A second,

mutually exclusive interaction with RPA is also described, which could indicate a regulatory role for the IDR requiring molecular hand-off.

The paper is a thorough exploration of the dynamics of interaction between the RECQ4 IDR domain and various forms of G4-quadruplexes. However, we have one major and some minor concerns/questions about the paper, detailed below. The materials and methods section requires more elaboration.

Major concern:

Figure 2f and Lines 130-137, Page 4: The authors state that the RECQ4 5E charge reversal mutant does not bind to RPA32C. While Supplementary Fig 3 does provide evidence of this through NMR, the IP panel of Fig.2f shows binding with the 5E mutant at the initial time-point (synchronized panel 0). The authors should address this contradictory data.

In our *in vitro* NMR-based approach, we tested for specific interactions between the Sld2-like region of RECQ4 (aa 1-400) and the well-known small protein-binding domains of RPA, RPA70N and RPA32C. We demonstrated a specific interaction between RSM (RECQ4 348-388) and the RPA32C domain (7 kDa) in this setup. This interaction is abolished *in vitro* with RECQ4-5E charge reversal mutant. We then tested for this 5E charge reversal mutant in the co-IP experiment between full-length RECQ4 protein (aa 1-1204) and full-length RPA heterotrimer (120 kDa) probing with RPA32 antibody. Our data clearly demonstrate that the 5E mutation significantly attenuates the interaction between the full-length proteins even at 0-hour release of G1/S synchronisation and is more evident when cells further progress in the S phase (2 hr G1/S release). The residual interaction between full-length RECQ4-5E and full-length RPA could be due to other domains of RECQ4 and/or RPA contributing to the interaction besides the RSM-RPA32C interface. To be consistent with our observations and avoid any contradictory interpretation, we now state in L147-L148 of the revised manuscript that “the basic patch of RSM motif is the principal determinant of RPA-RECQ4 interaction (Fig. 2f)”. Additionally, we also included a sentence in discussion (L434-L436) stating: “However, we cannot exclude the possibility that the RSM-RPA32C interaction may be stabilized by other regions in the context of the RPA heterotrimer.”

Minor concerns/suggestions:

1) Page 2, line 56-57: The authors state that ‘RECQ4 has a unique domain organization. It lacks the RQC and HRDC domains required for G4 unwinding by other members of the family.’. Fig. 1a shows that RQC is present in every RECQ protein but HRDC is not, so this sentence needs to be clarified – are both domains essential?

Based on the mechanisms reported for bacterial RECQ, BLM, and WRN helicase, it appears that RQC is required for G4 recognition in all three proteins. In contrast, the HRDC domain, present only in BLM and WRN, exhibits an auxiliary role by binding ssDNA flanking segments. To avoid confusion, we now state only the RQC domain in L64-L65 of the revised text. This is different for RECQ4, which lacks RQC and, in contrast to other members, contains the Sld2-like domain.

2) Page 2, line 64-65: ‘The Sld2-like region is largely unstructured (Fig1 and supplementary Fig1a)’. It is not immediately clear how the conclusion of the region being unstructured is being drawn from these figures. Are the authors referring to the NMR chemical shifts? More elaboration is required in the text.

We have prepared a new “Supplementary Fig. 1” cited in L73 of the revised manuscript, where sequence analysis and AlphaFold2 structure prediction demonstrate the disordered nature of the RECQ4 Sld2-like region.

3) Page 4, line 117-129:

a. Line 119 – please edit to ‘this allowed us to further map’

Thank you, it has been corrected in L127 of the revised manuscript.

b. In Fig 2c please provide information on the multiple sequence alignment used to generate this data, and add the reference of the software used for analysis

The following sentence describes the new “Supplementary Fig. 4c: “Conservation score from multiple sequence alignment of RECQ4(322-400) using 28 RECQ4 protein sequences was calculated using the ConSurf server [3]. Alignment of 28 sequences from UniProt covering the Euteleostomi taxonomic range was generated by MUSCLE1 v3.8.3 [4]. Sequence alignment is available in the Source Data of the article.” The corresponding references are listed in Supplementary References.

4) Page 5, line 162: The Supplementary table mentioned in the text is missing

We apologize for this mistake. The table was accidentally removed during the manuscript submission. It is now part of the Supplement and cited as “Supplementary Table 1” in L176 of the revised manuscript.

5) Page 7, line 223: What is the value of the Smolochowski constant being considered here, the size of the model being used, and does the model account for crowding?

We have considered only the most conservative diffusion-limited association rate, ca. $10^9 \text{ M}^{-1} \text{ s}^{-1}$, in which orientational constraints are neglected (which would decrease the limit by multiple orders of magnitude). This has been clarified in the revised text (L245-L246).

6) Page 8, line 248: A figure depicting the structures of T59-2T and HT could be a helpful addition here

The G4 structures are depicted in the new “Supplementary Fig. 12” of the revised manuscript (cited in L269).

7) Page 9, line 291: Was a control experiment measuring the CD spectrum of just the RSM peptide done, and could that be added to the panel showing the comparative CD spectra to ensure that the changes in the signal are only due to the interactions between RSM and the DNA and not due to the presence of RSM itself.

The free RSM peptide does not contribute any CD signal beyond 220 nm and thus does not interfere with the unique CD spectral signatures of G-quadruplexes. This is shown in the new “Supplementary Fig. 16 a” cited in L313 of the revised manuscript, which states that: “sRSM (aa 358-388) did not interfere with the G4 CD spectral features (Supplementary Fig. 16 a)”

8) Materials and Methods section:

a. Please add accession codes for proteins used

Uniprot accession codes have been added for all proteins used (L517-L518 and L572-L573 of the revised manuscript).

b. Page 12, line 387 and line 420: Please provide plasmid maps/ more information on cloning sites

The vector maps are shown in the Source Data of the article. The revised manuscript mentions the cloning sites in L470, L473, L519, and L531-L532.

c. Page 12, line 412: Please provide details of antibodies used – company, catalog number etc. Also details of the chemiluminescence using HRP substrate – if this was a kit, please mention which one.

We have expanded the first five sections of the Methods, L463-L513 of the revised manuscript, to describe in more detail the procedures followed in the cell-based experiments, as requested.

d. Page 13, line 439: Please provide primer sequences for cloning and expression protocol for RSM.

The primer sequences are displayed in the new “Supplementary Table 5”. The expression protocols for all proteins are described in the Methods, Protein preparation section, L534-L574 of the revised manuscript.

e. Page 13, line 457: Were the DNA substrates checked for depurination after heating for 10 min at 95C?

We did not check for depurination, but the imino signals in 1D NMR in all our experiments with different DNA substrates confirm the quality and proper folding of the DNA molecules. In the case of depurination, the NMR signals would change, and we would not be able to reproduce the NMR peaks and their assignments reported in the original studies of the G-quadruplexes.

f. Page 15, line 498: What was the label used for DNA?

The DNAs used for EMSA (L658-L680 of the revised manuscript) were FITC- or Cy3- labeled. The DNA labeling position is stated in the new "Supplementary Table 1", along with the sequences of DNA substrates used in the study.

9) Supplementary Figures:

a. Supp. Fig.10: Please include the polydispersity index of the peaks

After carefully inspecting our experimental procedures, we realized that the person who performed the DLS experiments was unintentionally centrifuging the samples and collecting the supernatant for the measurements before the analysis. This most likely resulted in the removal of droplets and monitoring only the small particles in the solution (RSM-G4 complex in dilute phase and any excess of free RSM or G4) and misinterpretation of the DLS data. We are grateful to the reviewers for pointing out the phase separation that allowed us to re-examine our data and perform additional controls. The DLS data are removed from the manuscript.

b. Supp. Fig.11, panel e: The SD is not present or not visible.

The experiment in new "Supplementary Fig. 20" was performed once to titrate the amount of ScPif1 used in the new "Fig. 8c". In the legend of the new "Supplementary Fig. 20", we state that quantification in panel (e) is based on the single experiment shown in panel (c).

c. Supp. Fig.12: Are the quantification panels an average of two runs? Please clarify.

Quantification was based on the single experiments shown in the new "Supplementary Fig. 21a, b" and this information is included in the legend.

REVIEWERS' COMMENTS

Reviewer #1 (Remarks to the Author):

The authors have effectively addressed the points raised by this reviewer in the revised version of the manuscript. As a result, I have no additional comments to provide.

Reviewer #2 (Remarks to the Author):

All issues pointed by the reviewers were adequately addressed on the manuscript was revised accordingly. I do not have new critiques.

Reviewer #4 (Remarks to the Author):

Although the in vitro LLPS assays in this manuscript are quite basic and kept minimal, they overall support the notion that RSM and G4 form complex coacervates and mediated by their electrostatic interactions.

Regarding the use of ThT to probe the G4 structure in Figure 7, the data are consistent with the retention of G4 structure in the coacervate phase. However, it should be pointed out that ThT probes not only G4 structure, but also (and very commonly used for) beta-sheet rich structures, which are found in amyloid fibrils. Liquid-like condensates can transition to states that may partially contain beta sheet-rich amyloid structures under certain in vitro conditions, and show increase in ThT staining signals. See PMID: 32661370. The authors thus should not exclude this possibility in their interpretation and discussion. It may help explain "Intriguingly, a similar increase in fluorescence was observed upon induction of coacervation by adding RSM" (Line 382).

Point-by-point response

We would like to thank the reviewers for endorsing our manuscript for publication. Below we address their final comments.

Reviewer #1 (Remarks to the Author):

The authors have effectively addressed the points raised by this reviewer in the revised version of the manuscript. As a result, I have no additional comments to provide.

Reviewer #2 (Remarks to the Author):

All issues pointed by the reviewers were adequately addressed and the manuscript was revised accordingly. I do not have new critiques.

We're pleased to hear that we have successfully addressed the points raised by the reviewers. Comments from both reviewers have been instrumental in improving the quality of this work.

Reviewer #4 (Remarks to the Author):

Although the in vitro LLPS assays in this manuscript are quite basic and kept minimal, they overall support the notion that RSM and G4 form complex coacervates and mediated by their electrostatic interactions. Regarding the use of ThT to probe the G4 structure in Figure 7, the data are consistent with the retention of G4 structure in the coacervate phase. However, it should be pointed out that ThT probes not only G4 structure, but also (and very commonly used for) beta-sheet rich structures, which are found in amyloid fibrils. Liquid-like condensates can transition to states that may partially contain beta sheet-rich amyloid structures under certain in vitro conditions, and show increase in ThT staining signals. See PMID: 32661370. The authors thus should not exclude this possibility in their interpretation and discussion. It may help explain “Intriguingly, a similar increase in fluorescence was observed upon induction of coacervation by adding RSM” (Line 382).

We would like to thank the reviewer for acknowledging that our data support RSM-G4 coacervation. In L385-387 we have added the following sentence stating that ThT signal may additionally arise from RSM fibrillation in the droplets: “Although we cannot exclude the possibility that ThT signal may partially result from RSM amyloid structures formed in the droplets¹²² ...” and provide the suggested reference.

Additionally, to reinforce that G4 retains its molecular structure in the coacervate phase, we have conducted an experiment using another G4 ligand, IMT (PMID: 30085206) and present the results in **Figure A** bellow. This experiment demonstrates IMT's staining of G4-induced droplets while showing no signal in the presence of ssDNA, providing further evidence for specific G4 staining. However, the status of beta-sheet staining by IMT is not known.

Figure A: Phase separation microscopy of sRSM (20 μM) mixed with T95-2T G4 or ssDNA-18mer (20 μM). IMT was added to a final concentration of 20 μM and the IMT fluorescence signal in the droplets was analysed by fluorescent microscopy. In all images scale bar corresponds to 5 μm .